# Mitigating Hallucination in Vision-Language Model with Depth and Spatial-aware Key-Value Refinement

**Gusang Lee**[1], **Soohyun Kim**[1], **Donghoon Kim**[1], **Kyuhong Shim**[2*], **Byonghyo Shim**[1*]

[1]Seoul National University [2]Sungkyunkwan University

{gslee, soohyunkim, dhkim, bshim}@islab.snu.ac.kr; khshim@skku.edu

## Abstract

Large vision–language models (VLMs) deliver state-of-the-art results on a wide range of multimodal tasks, yet they remain prone to visual hallucinations, producing content that is not grounded in the input image. Despite progress with visual supervision, reinforcement learning, and post-hoc attention reshaping, the representational origins of hallucinations remain unclear. Our study reveals that successful grounding emerges when adjacent visual tokens exhibit coherent alignment, while hallucinations arise when key vectors scatter isotropically, weakening cross-modal attention and blurring object boundaries. Building on this insight, we propose Depth and Spatial aware Cache Refinement (DSCR), a lightweight and training-free method that augments the Transformer's key-value (KV) cache with depth cues and 2D spatial proximity. DSCR clusters vectors within objects and separates those across surfaces, guiding attention toward relevant regions without any fine-tuning. Comprehensive evaluations show that DSCR consistently reduces hallucinations, delivering up to 41.6% accuracy gains across MME, POPE, RePOPE, CHAIR, and a new depth-sensitive benchmark. Our findings highlight KV-coherence as a core factor behind hallucinations and demonstrate a practical, model-agnostic solution for enhancing VLM reliability.

## 1 Introduction

In recent years, we have witnessed remarkable advances in large vision-language models (VLMs), such as GPT-5, Claude-4, and Gemini-2.5 (OpenAI, 2025; Anthropic, 2025; Gemini Team, 2025). VLMs are widely used in vision-related tasks, including AR solutions (e.g., real-time navigation), VLM agent-driven automation (e.g., smart shopping assistants), robot control, and visual content generation (Xiu et al., 2025; Song et al., 2025; Niu et al., 2024; Xu et al., 2024; Yuan et al., 2024; Li et al., 2023b; Guo et al., 2025; Ge et al., 2025).

One well-known drawback that undermines the reliability of VLMs is visual hallucination. By visual hallucination, we mean the phenomenon of generating content that is not grounded in input visual information (Li et al., 2023c; Huang et al., 2025; Sahoo et al., 2024). Recent studies show that visual hallucinations emerge when the visual encoder fails to extract sufficient information from the input image to answer the natural language query. In this scenario, VLM falls back on the linguistic priors it learned from large-scale text corpora rather than visual input data. As a result, VLM would describe objects that do not exist, assign incorrect attributes to visible objects, or misinterpret spatial relationships within the scene.

Over the years, various approaches have been proposed to mitigate visual hallucination phenomena. Prior work has explored three main strategies: augmenting training with extra visual supervision (e.g., bounding boxes or segmentation masks) to improve grounding (Jain et al., 2024; Wan et al., 2025), using reinforcement learning based reward modeling to align outputs with human judgments (Sun et al., 2023), and applying lightweight, training-free techniques that reshape attention patterns or filter out low-confidence predictions without touching model weights (Leng et al., 2024b; Huang et al., 2024; Chen et al., 2024b; Wang et al., 2025; An et al., 2025). To some extent, these approaches

---
*Corresponding authors

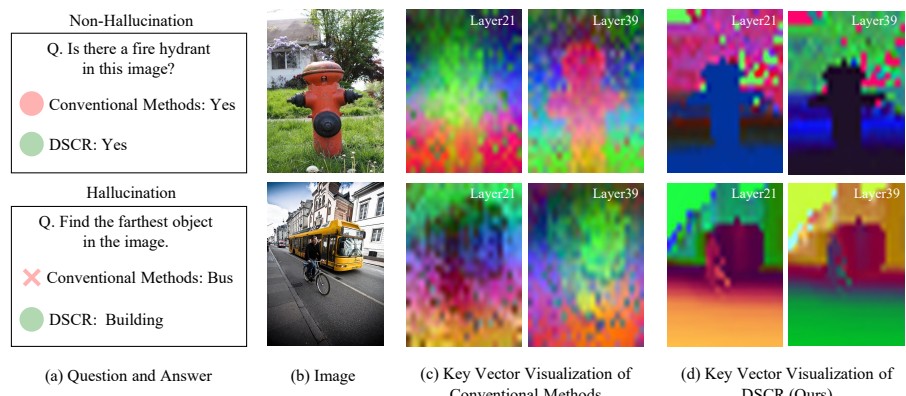

Figure 1: Visualization of non-hallucination (top) and hallucination (bottom) cases with key vector outputs. (a) Example questions and answers. (b) Input images. (c) Key vector visualizations from conventional methods. (d) Key vector visualizations after applying DSCR, showing more structured and spatially aligned features, especially in hallucination cases.

have shown benefits; however, a clear and shared explanation of why hallucinations emerge remains unresolved. In particular, it remains unclear how the model's internal representations change when exposed to a specific image and question, ultimately leading to hallucinations.

To investigate the fundamental reason why VLMs struggle to retrieve query-relevant information from visual inputs, we applied principal component analysis (PCA) to the transformer's key vectors under both hallucinated and non-hallucinated conditions (see Section 2.4). As shown in Figure 1(c) and (d), when hallucination does not occur, the key vectors of the spatially adjacent patches are well aligned. This spatial coherence becomes even stronger in deeper layers. In contrast, when hallucination occurs, the vectors scatter nearly isotropically, blurring the boundaries of the object. Our analysis suggests that both phenomena arise from a fundamental representational mechanism: a loss of coherence among key vectors, which leads the vectors to scatter and blur object boundaries, thereby preventing the cross-attention mechanisms from anchoring onto visual representations. Conversely, when key vectors maintain coherent alignment across adjacent patches, they facilitate robust cross-modal attention flows that reliably transmit visual information to the language model.

Building on our PCA analysis of key vector coherence, we propose a lightweight and training-free mechanism that steers cross-attention to the most relevant image regions by refining the model's internal KV cache. We call this depth and spatial aware cache refinement (DSCR): it injects both relative depth and 2D spatial proximity cues directly into every key vector, with no fine-tuning needed. Depth provides true 3D structure, sharpening object edges at depth discontinuities, separating overlapping foreground/background surfaces, and down-weighting occlusions. In parallel, 2D spatial proximity reinforces local context, ensuring that immediately neighboring patches (which often share texture or semantics) remain tightly clustered in representation space. By combining these geometric and planar signals, DSCR guides key vectors on the same object to form a coherent group and pushes apart those on different surfaces or distant in the image plane. The result is a geometry and locality-aware attention pattern that reduces spurious hallucinations without additional training.

Our contributions are summarized as follows:

- **Model agnostic KV refinement**: We introduce a plug-and-play technique that boosts the similarity of key vectors for spatially adjacent tokens belonging to the same object, enabling VLMs to better capture core visual structures without any finetuning.

- **First analysis of KV-coherence vs. hallucination**: We uncover how the breaks in neighboring-key similarity trigger visual hallucinations using PCA-based visualizations, attention-score diagnostics against query tokens, and end-to-end tests on our new depth-focused hallucination benchmark.

- **Broad benchmark gains**: Across MME, POPE, RePOPE, CHAIR, and our novel benchmark, DSCR delivers up to 41.6% improved accuracy. Moreover, DSCR can be seamlessly combined with existing mitigation strategies, providing complementary improvements.

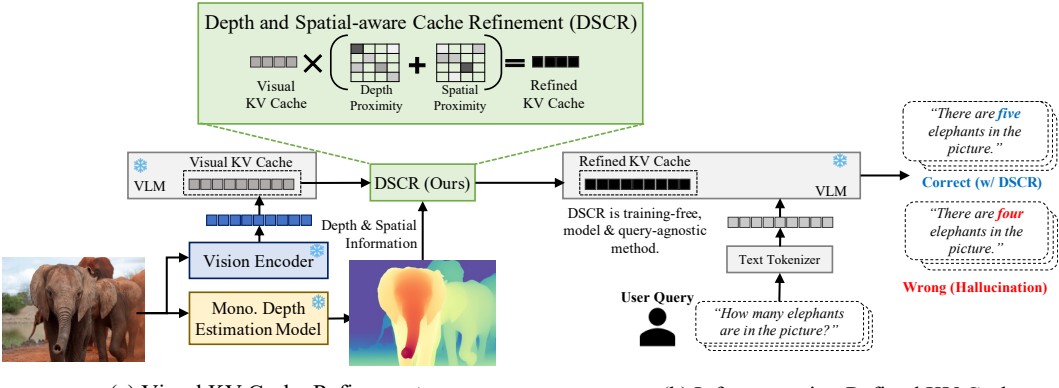

(a) Visual KV Cache Refinement     (b) Inference using Refined KV Cache

Figure 2: Illustration of the proposed **DSCR**. Extracted visual tokens first pass through the frozen VLM. Then, the corresponding KV cache is refined using the depth and spatial relationship between tokens. By this procedure, DSCR establishes a strong association between relevant visual tokens in the attention blocks, mitigating hallucinations in the VLMs. The VLMs with DSCR produce the accurate answer (blue), unlike the original VLM (red). Note that DSCR is training-free, model-invariant, and query-agnostic.

## 2 DEPTH AND SPATIAL AWARE KEY-VALUE CACHE REFINEMENT

### 2.1 NOTATION AND OVERVIEW

**Notation.** The token representation and notation used in VLMs are defined as follows. Given an RGB image $I \in \mathbb{R}^{H \times W \times 3}$, the vision encoder divides the image into non-overlapping patches and extracts $N$ visual tokens. These tokens are passed as a sequence to the language model, which then consists of $L$ transformer layers, each with $H$ attention heads. Each head maintains a KV cache, denoted as $\mathbf{K}^I = [\mathbf{k}_1^I, \ldots, \mathbf{k}_N^I]$ and $\mathbf{V}^I = [\mathbf{v}_1^I, \ldots, \mathbf{v}_N^I]$, where $(\mathbf{k}_i^I, \mathbf{v}_i^I) \in \mathbb{R}^{d_h}$ represents the key and value vectors for the $i$-th token, and $d_h$ is the head dimension.

**Overview.** Using DSCR, we can improve both the similarity between key vectors of neighboring image tokens and the model's ability to identify object boundaries. This implies that the refined model can more accurately attend to query-relevant visual regions during decoding. Figure 2 illustrates where DSCR is applied in the model architecture. It refines the visual KV cache prior to decoding by leveraging auxiliary depth and spatial information. Importantly, DSCR does not require model weight updates or fine-tuning. It operates exclusively at inference time, increasing the similarity among visual token representations in a lightweight, modular manner.

### 2.2 THEORETICAL RATIONALE: DEPTH AND SPATIAL PRIORS

Our design is grounded in classical image priors and recent graph-signal theory. Natural images are characterized by dominant low-frequency energy and strong covariance among neighboring patches. This observation motivates locality-aware Vision Transformer (ViT) variants such as LocalViT (Li et al., 2021) and SATA (Nikzad et al., 2025). Depth adds a complementary geometric prior: pixels with nearly identical disparity almost always lie on the same physical surface, while sharp depth discontinuities coincide with true object edges (Tomasi & Manduchi, 1999). We encode both priors by re-weighting key–value pairs in the cache, yielding an edge-aware attention operator that enforces local smoothness (via 2D spatial proximity) and respects object boundaries (via depth). Mathematically, this acts as a graph-Laplacian smoothing term on the key–vector graph, clustering tokens on the same surface into a low-frequency subspace and dispersing those across depth gaps. As a result, it suppresses high-frequency noise and amplifies reliable local evidence.

Recent analyses show that hallucinations often arise when models over-rely on global features and neglect local cues; explicitly combining global and local attention reduces such errors (Xing et al., 2024; An et al., 2025), and our depth-guided smoothing achieves this integration implicitly. Anal-

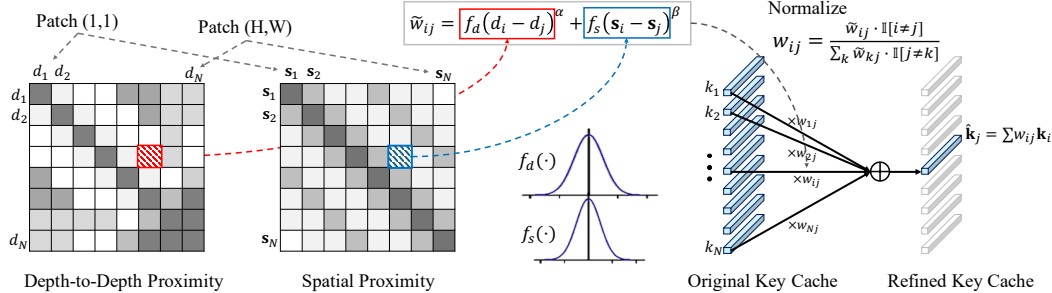

Figure 3: Details of the DSCR's key cache refinement process. Depth-to-depth and spatial proximity maps show the difference in depth values and distances between image patch pairs, with darker shades indicating smaller differences. Using the importance weights ($w_{ij}$) derived from the proximity scores, refined cache entries are computed as a weighted sum of the original ones.

ogously, DFormerv2 (Yin et al., 2025) inserts depth-based geometry self-attention into a ViT backbone, achieving SOTA on NYUD-v2 and SUN RGB-D datasets, demonstrating the practical impact of depth-guided coupling.

## 2.3 DEPTH AND SPATIAL-AWARE CACHE REFINEMENT

**Depth Estimation.** To extract the depth map $\mathbf{D} \in \mathbb{R}^{H \times W}$ from the RGB image $I$, we use an off-the-shelf monotonic depth estimation (MDE) model. The depth map is then min-max normalized to scale the depth values to $[0, 1]$. To match the resolution of the vision encoder's output, we resize the depth map and obtain depth values $\{d_1, d_2, \cdots, d_N\}$. Each $d_i$ represents the depth value of the $i$-th image patch, which corresponds to the key-value (KV) cache entries $\mathbf{k}_i^I$ and $\mathbf{v}_i^I$.

**Depth-to-Depth Proximity.** To identify the relationship between the $i$ and $j$-th image patches, we measure the depth-to-depth proximity using the Gaussian kernel:

$$f_d(d_i - d_j) = \exp\left(-\frac{(d_i - d_j)^2}{2\sigma_d^2}\right), \tag{1}$$

where $\sigma_d$ is a hyperparameter to control the width of the Gaussian function. One can notice that the proximity score is higher when the depth difference is small, indicating higher similarity for patches with similar depths.

**Spatial Proximity.** Similarly, we compute the spatial proximity between $i$ and $j$-th image patches using a Gaussian function of the Euclidean distance:

$$f_s(\mathbf{s}_i - \mathbf{s}_j) = \exp\left(-\frac{\|\mathbf{s}_i - \mathbf{s}_j\|_2^2}{2\sigma_s^2}\right), \tag{2}$$

where $\mathbf{s}_i$ and $\mathbf{s}_j$ are 2D pixel coordinates of each patch. Note that the proximity score increases if the two patches are close in an image.

**Total Proximity and Importance Weight.** The total proximity between $i$-th and $j$-th image patches is computed as:

$$\tilde{w}_{ij} = f_d(d_i - d_j)^\alpha + f_s(\mathbf{s}_i - \mathbf{s}_j)^\beta, \tag{3}$$

where $\alpha, \beta$ are hyperparameters controlling the contribution of depth and spatial proximity, respectively. To ensure that each token is refined based solely on its neighbors, we set the self-proximity term $\tilde{w}_{jj}$ to zero by masking out the diagonal of the proximity matrix. We normalize this proximity score to obtain the relative importance weight as

$$w_{ij} = \frac{\tilde{w}_{ij} \cdot \mathbb{I}[i \neq j]}{\sum_k \tilde{w}_{kj} \cdot \mathbb{I}[k \neq j]}, \tag{4}$$

where $\mathbb{I}[\cdot]$ is the indicator function whose value is one if the condition is satisfied, otherwise zero. For the hyperparameters, we used fixed values for $\sigma_d = 0.6$, $\sigma_s = 0.6$, $\alpha = 0.6$, $\beta = 0.8$.

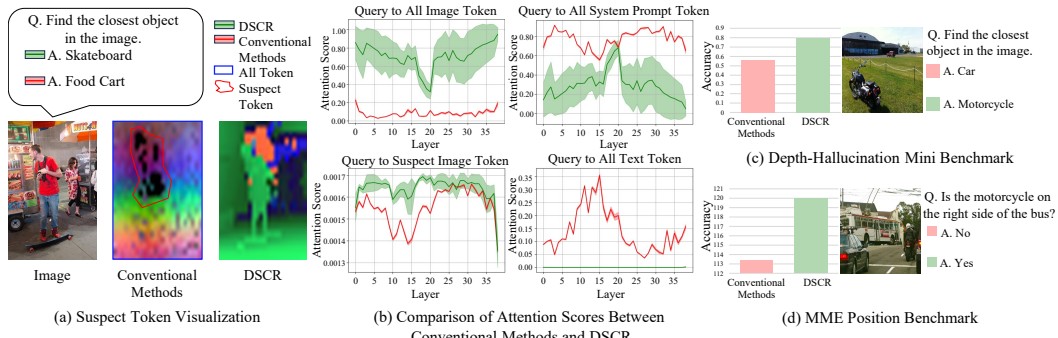

Figure 4: Analysis of hallucination suppression using DSCR. (a) Visualization of a hallucination case where DSCR reduces incorrect predictions by suppressing suspect tokens and enhancing the spatial structure of key vectors. (b) Layer-wise attention scores comparing Conventional Methods and DSCR across image, suspect, prompt, and text tokens. (c) Accuracy comparisons from the Hallucination-Depth Mini Benchmark, comparing the Conventional Methods with depth-refined KV cache. (d) Results on MME where spatial refinement is applied to KV caches.

**Key-Value Cache Refinement.** Using the relative importance weight, we update each Key and Value entry by computing a weighted sum of all Key and Value entries, respectively. For the $j$-th entry, the refinement proceeds as follows:

$$\hat{\mathbf{k}}_j^I = \sum_i w_{ij}\mathbf{k}_i^I, \quad \hat{\mathbf{v}}_j^I = \sum_i w_{ij}\mathbf{v}_i^I. \tag{5}$$

The same weights are applied across the selected Transformer layers and attention heads. In practice, this weighted sum can be performed by a single inter-tensor multiplication, ensuring that the DSCR computation remains highly efficient and negligible. The refined KV cache replaces the original KV cache before the text generation. Note that this modification is applied only to the KV cache entries corresponding to visual tokens.

We highlight that the entire DSCR process is training-free, model-agnostic, and query-agnostic. Furthermore, our DSCR proposes the novel concept of refining KV caches, which has not been previously explored in hallucination prevention within both large language model (LLM) and VLM literature.

## 2.4 Comprehensive Analysis of Hallucination in LVLM

Figure 4 illustrates the superiority of DSCR from three perspectives: key vector visualization, attention distribution, and benchmark-level performance gains reflecting improved utilization of depth and spatial information.

**Key Vector Visualization.** The visualizations shown in Figure 4(a) include the input image, the key vectors from the conventional method, and the key vectors after applying DSCR. Each key vector $\mathbf{k}_i$ is projected into 3D via PCA for RGB visualization. The process is defined as:

$$k_i^{\text{RGB}} = \text{Norm}\left(\text{PCA}_3(\mathbf{k}_i)\right) \in \mathbb{R}^3, \tag{6}$$

where $\text{PCA}_3(\cdot)$ projects the key vector into a 3-dimensional space, and $\text{Norm}(\cdot)$ denotes channel-wise min-max normalization.

In VLMs, key vectors are not structurally aligned, and neighboring tokens often exhibit disjoint directions, particularly in hallucination cases. Notably, in the suspect token region (highlighted in red), a sharp deviation in vector direction is observed (black region), which may undermine the model's ability to correctly interpret the visual input. In contrast, DSCR leads to smooth alignment among nearby tokens, with clearer object boundaries and the disappearance of suspect tokens. This behavior results from DSCR's use of depth and spatial proximity to align KV vectors for nearby patches and distinguish objects based on relative depth differences.

Table 1: Detailed evaluation results on MME hallucination subset. Best results in **bold**.

| Model | Metric | Baseline | VCD | OPERA | HALC | DAMO | AGLA | DSCR (Ours) | VCD +DSCR | OPERA +DSCR | HALC +DSCR | DAMO +DSCR | AGLA +DSCR |
|---|---|---|---|---|---|---|---|---|---|---|---|---|---|
| LLaVA-1.5 | Existence | 190.00 | 190.00 | **195.00** | 190.00 | 190.00 | 190.00 | **195.00** | **195.00** | **195.00** | **195.00** | 190.00 | 190.00 |
| | Count | 143.33 | 143.33 | **160.00** | 153.33 | 148.33 | 135.00 | **160.00** | 143.33 | **160.00** | **160.00** | 155.00 | 155.00 |
| | Position | **120.00** | **120.00** | **120.00** | **120.00** | 105.00 | **120.00** | **120.00** | 115.00 | **120.00** | **120.00** | 110.00 | **120.00** |
| | Color | 165.00 | 165.00 | 165.00 | 170.00 | 160.00 | 165.00 | **175.00** | 170.00 | **175.00** | **175.00** | 165.00 | **175.00** |
| | OCR | 117.50 | 117.50 | 117.50 | 125.00 | 117.50 | 117.50 | **140.00** | 110.00 | **140.00** | **140.00** | 117.50 | **140.00** |
| | Posters | 156.85 | 156.85 | 157.19 | 141.10 | 151.37 | 162.67 | 135.96 | 155.14 | 135.37 | 135.96 | 148.97 | 151.37 |
| | **Total** | 892.68 | 892.68 | 914.69 | 899.43 | 872.20 | 890.17 | 925.96 | 888.47 | 925.27 | 925.96 | 886.47 | **931.37** |
| LLaVA-1.6 | Existence | **180.00** | **180.00** | **180.00** | 175.00 | **180.00** | 175.00 | **180.00** | **180.00** | 170.00 | **180.00** | **180.00** | 170.00 |
| | Count | 153.33 | 138.33 | **158.33** | 141.67 | 150.00 | 146.67 | 156.67 | 136.67 | 135.00 | 156.67 | 155.00 | 143.33 |
| | Position | 93.33 | 93.33 | 98.33 | 101.67 | 96.67 | 91.67 | 105.00 | 110.00 | 105.00 | 105.00 | 101.67 | **113.33** |
| | Color | **180.00** | **180.00** | **180.00** | 155.00 | 175.00 | 175.00 | 175.00 | 175.00 | 175.00 | 175.00 | 155.00 | 150.00 |
| | OCR | 132.50 | **147.50** | 132.50 | 132.50 | 117.50 | 140.00 | 132.50 | **147.50** | **147.50** | 132.50 | 132.50 | **147.50** |
| | Posters | 150.34 | 147.60 | 150.34 | 142.47 | 151.37 | 157.53 | 152.40 | 148.63 | 132.88 | 152.38 | 142.47 | 144.52 |
| | **Total** | 889.51 | 886.77 | 899.51 | 848.30 | 870.54 | 885.87 | **901.56** | 897.80 | 835.38 | **901.56** | 866.63 | 868.69 |
| Qwen-VL | Existence | 170.00 | 165.00 | 170.00 | 170.00 | 160.00 | 170.00 | 175.00 | 165.00 | 165.00 | **185.00** | 153.33 | 175.00 |
| | Count | 145.00 | 145.00 | 150.00 | 145.00 | 150.00 | 145.00 | **155.00** | 145.00 | 150.00 | 145.00 | **155.00** | 145.00 |
| | Position | 98.33 | 98.33 | 98.33 | 98.33 | 108.33 | 93.33 | 103.33 | 98.33 | 101.67 | 108.33 | **121.67** | 96.67 |
| | Color | 180.00 | 180.00 | 180.00 | 180.00 | **185.00** | 180.00 | **185.00** | 180.00 | 180.00 | **185.00** | 180.00 | 175.00 |
| | OCR | 87.50 | 110.00 | 87.50 | 87.50 | **110.00** | 87.50 | 87.50 | 102.50 | 87.50 | 80.00 | **110.00** | 80.00 |
| | Posters | 144.18 | 146.23 | 144.52 | 144.18 | 152.05 | 146.58 | 166.78 | 148.92 | 160.62 | **178.42** | 145.55 | 154.79 |
| | **Total** | 825.01 | 844.57 | 830.35 | 825.01 | 865.39 | 822.41 | 872.61 | 839.81 | 844.78 | **881.76** | 865.55 | 826.46 |
| Qwen2.5-VL | Existence | **195.00** | **195.00** | 190.00 | 185.00 | 190.00 | 190.00 | 190.00 | **195.00** | 185.00 | 190.00 | 190.00 | 190.00 |
| | Count | 160.00 | 165.00 | 165.00 | 135.00 | 160.00 | 165.00 | 165.00 | **170.00** | 165.00 | 165.00 | 165.00 | 165.00 |
| | Position | 160.00 | 160.00 | 160.00 | 153.33 | 160.00 | **165.00** | 155.00 | 160.00 | 155.00 | 155.00 | 155.00 | **165.00** |
| | Color | 185.00 | 190.00 | 185.00 | **195.00** | 185.00 | 190.00 | 185.00 | 190.00 | 190.00 | 185.00 | 185.00 | 190.00 |
| | OCR | **177.50** | 155.00 | 170.00 | 155.00 | 170.00 | 162.50 | **177.50** | 162.50 | **177.50** | **177.50** | **177.50** | 162.50 |
| | Posters | 165.41 | 167.12 | 165.41 | 164.73 | 167.47 | 167.81 | 167.47 | 168.49 | 166.10 | **169.52** | 168.49 | 167.81 |
| | **Total** | 1042.91 | 1032.12 | 1035.41 | 988.06 | 1032.47 | 1040.31 | 1039.97 | **1045.99** | 1043.60 | 1042.02 | 1040.99 | 1040.31 |

**Attention Score Analysis.** The effect of DSCR on attention distribution is shown in Figure 4(b). The three plots show layer-wise attention scores from the query to all image tokens, system prompt tokens, and text tokens. The model often exhibits little attention to image tokens regardless of the question, instead focusing heavily on system prompts and textual priors. In contrast, DSCR consistently increases attention to relevant image tokens across different inputs while reducing reliance on text and prompt tokens. This indicates that DSCR shifts the model toward visual-grounded reasoning. The bottom-left plot in Figure 4(b) shows the average attention scores assigned to image tokens at the suspect token indices for each query. Without DSCR, these positions exhibit notably low attention scores, suggesting that suspect tokens interfere with the model's ability to attend to relevant visual information. In contrast, DSCR mitigates this issue, and higher attention scores are consistently maintained at those positions.

**Depth & Position Hallucination Benchmark for Occlusion and Similar-Depth Scenarios.** In typical datasets, occlusion and boundary pixels occupy less than 4% of the scene, making depth discontinuities relatively rare (Birchfield & Tomasi, 1999). In contrast, to demonstrate the robustness of DSCR even under challenging conditions, such as occlusion boundaries and semantically distinct objects located at similar depths, we constructed a depth hallucination mini-benchmark by selecting 50 images containing heavily overlapping objects with comparable depth values. For each image, we designed four questions asking about the closest, farthest, physically smallest, and physically largest objects, in order to evaluate the model's ability to reason about relative depth and real-world object size. The position hallucination mini-benchmark is built from the position subset of the MME (Fu et al., 2023) dataset to assess spatial relationship reasoning. As shown in Figure 4(c), DSCR improves accuracy on the depth hallucination mini-benchmark by 41.4%, indicating better utilization of depth information. Additionally, Figure 4(d) demonstrates a 6% gain on the position hallucination mini-benchmark, confirming that spatial proximity contributes to more accurate spatial reasoning. See Appendix for more details about our mini-benchmark.

Table 2: Evaluation results on POPE (GQA) and RePOPE (MSCOCO) datasets.

| Setting | Model | w/DSCR | POPE (GQA) | | | | RePOPE (MSCOCO) | | | |
|---------|-------|--------|------|-------|------|------|------|-------|------|------|
| | | | Acc. | Prec. | Rec. | F1 | Acc. | Prec. | Rec. | F1 |
| Random | LLaVA-1.5 | × | 0.90 | 0.92 | **0.87** | 0.89 | 0.92 | 0.93 | 0.88 | 0.90 |
| | | ✓ | 0.90 | **0.93** | 0.85 | 0.89 | 0.92 | 0.95 | 0.87 | 0.90 |
| | Qwen-VL | × | 0.84 | 0.84 | **0.83** | 0.84 | 0.92 | 0.92 | 0.88 | 0.90 |
| | | ✓ | **0.85** | **0.91** | 0.77 | 0.84 | 91 | **0.93** | 0.87 | 0.90 |
| | mPLUG-Owl2 | × | **0.85** | 0.91 | **0.77** | **0.84** | 0.63 | 0.62 | 0.87 | 0.70 |
| | | ✓ | 0.84 | **0.93** | 0.75 | 0.83 | **0.70** | **0.68** | **0.89** | **0.78** |
| Popular | LLaVA-1.5 | × | 0.86 | 0.84 | **0.87** | 0.86 | 0.91 | 0.92 | 0.86 | 0.89 |
| | | ✓ | **0.87** | **0.87** | 0.85 | 0.86 | **0.90** | **0.93** | **0.85** | **0.89** |
| | Qwen-VL | × | 0.72 | 0.68 | **0.83** | 0.75 | 0.89 | 0.87 | 0.88 | 0.88 |
| | | ✓ | **0.80** | **0.81** | 0.77 | **0.79** | **0.88** | **0.88** | **0.87** | **0.88** |
| | mPLUG-Owl2 | × | 0.79 | 0.80 | **0.77** | **0.79** | 0.60 | 0.55 | 0.87 | 0.68 |
| | | ✓ | 0.79 | **0.82** | 0.75 | 0.78 | **0.66** | **0.60** | **0.90** | **0.74** |
| Adversarial | LLaVA-1.5 | × | 0.81 | 0.78 | **0.87** | 0.82 | 0.89 | 0.87 | 0.87 | 0.87 |
| | | ✓ | **0.82** | **0.80** | 0.85 | 0.82 | **0.89** | **0.89** | **0.85** | **0.87** |
| | Qwen-VL | × | 0.75 | 0.72 | 0.77 | 0.77 | 0.89 | 0.87 | 0.88 | 0.88 |
| | | ✓ | **0.78** | **0.79** | 0.77 | **0.78** | **0.88** | **0.88** | **0.87** | **0.88** |
| | mPLUG-Owl2 | × | 0.77 | 0.76 | **0.77** | 0.77 | 0.58 | 0.53 | 0.85 | 0.66 |
| | | ✓ | 0.77 | **0.79** | 0.75 | 0.77 | **0.63** | **0.58** | **0.88** | **0.72** |

## 3 EXPERIMENT

### 3.1 HALLUCINATION MITIGATION

**MME Hallucination Subset.** Table 1 shows that DSCR achieves consistent performance improvements across various VLMs. The MME (Fu et al., 2023) total scores increase from 892.68 to 925.96 (+33.28 points) for LLaVA-1.5 (Liu et al., 2024c), from 889.51 to 901.56 (+12.05 points) for LLaVA-1.6 (Liu et al., 2024b), and from 825.01 to 872.61 (+47.60 points) for Qwen-VL (Bai et al., 2023).

Table 3: Evaluation results on the CHAIR dataset using LLaVA-1.5.

| Method | CHAIR$_S$ ↓ | CHAIR$_I$ ↓ | Recall ↑ | Avg. Len. |
|--------|-------------|-------------|----------|-----------|
| Baseline | 48.2 | 12.5 | 78.2 | 99.4 |
| VCD | 54.4 | 14.4 | **79.0** | 101.6 |
| OPERA | **39.2** | 11.6 | 73.5 | 83.9 |
| **DSCR** | **39.2** | **11.2** | 73.4 | 96.1 |

In comparison with prior methods on LLaVA-1.5, DSCR improves the total score from 914.69 to 925.96 (+11.27 points) over VCD (Leng et al., 2024b), from 872.20 to 925.96 (+53.76 points) over DAMO (Wang et al., 2025), from 899.43 to 925.96 (+26.53 points) over HALC (Chen et al., 2024b), and from 890.17 to 925.96 (+35.79 points) over AGLA (An et al., 2025). Furthermore, applying DSCR on top of existing inference-time methods consistently provides additional absolute gains in total score.

**POPE & RePOPE.** Table 2 demonstrates that DSCR consistently improves F1 scores on both the POPE (Li et al., 2023c) and RePOPE (Neuhaus & Hein, 2025) in terms of absolute points.

For LLaVA-1.5, DSCR improves the F1 score on POPE from 0.87 to 0.90 (+0.03) under the Random strategy, from 0.85 to 0.88 (+0.03) under the Popular strategy, and from 0.80 to 0.83 (+0.03) under the Adversarial strategy. On RePOPE, the F1 score increases from 0.72 to 0.80 (+0.08), from 0.70 to 0.75 (+0.05), and from 0.68 to 0.73 (+0.05) under the Random, Popular, and Adversarial settings, respectively. Additional POPE results on MSCOCO (Lin et al., 2014) and A-OKVQA (Schwenk et al., 2022), as well as experiments applying DSCR to existing inference-time methods such as VCD and OPERA to further improve F1 scores, are provided in the Appendix.

**CHAIR.** Table 3 presents evaluation results on the CHAIR (Rohrbach et al., 2018) benchmark, which measures object hallucination in image captioning. Compared to the baseline, VCD, and OPERA, DSCR achieves the lowest CHAIR$_I$ score (11.2) and matches the lowest CHAIR$_S$ score

Table 4: Evaluation results on AMBER dataset. A, P, R, F1 indicates accuracy, precision, recall, and F1-score, respectively. Please see the Appendix for more details.

| Method | Generative | | | | Discriminative | | | | Existence | | | | Relation | | | | Attribute | | | | State | | | | Number | | | |
|---|---|---|---|---|---|---|---|---|---|---|---|---|---|---|---|---|---|---|---|---|---|---|---|---|---|---|---|---|
| | CHAIR | Cover | Hal | Cog | A | P | R | F1 | A | P | R | F1 | A | P | R | F1 | A | P | R | F1 | A | P | R | F1 | A | P | R | F1 |
| Baseline | 6.7 | 51.3 | 31.1 | 3.5 | 71.5 | **96.0** | 59.5 | 73.5 | 66.0 | **100.0** | 66.0 | 79.5 | 70.3 | **93.7** | 30.2 | 45.7 | 75.3 | **90.8** | 56.4 | 69.6 | 71.7 | **88.5** | 50.0 | 63.9 | 79.9 | **93.0** | 64.6 | 76.2 |
| VCD | 7.9 | **52.0** | 34.9 | 3.7 | 71.0 | 95.0 | 59.5 | 73.2 | 64.3 | **100.0** | 64.3 | 78.2 | 70.0 | 90.6 | 30.6 | 45.7 | 75.6 | 89.0 | 58.4 | 70.5 | 72.7 | 86.4 | 53.8 | 66.3 | 78.9 | 92.0 | 63.2 | 74.9 |
| **DSCR** | **6.3** | 50.4 | **29.4** | **3.1** | **75.2** | 95.4 | **65.7** | **77.8** | **71.6** | **100.0** | **71.6** | **83.4** | **71.4** | 93.1 | **33.4** | **49.2** | **78.3** | 89.7 | **63.9** | **74.6** | **75.5** | 87.6 | **59.4** | **70.8** | **81.9** | 92.6 | **69.3** | **79.3** |

Figure 5: Visual question-answering examples, including image, depth, and query-to-image attention heatmaps before and after applying DSCR. Yellow parts indicate large attention probabilities.

(39.2). Although the recall is 73.4, DSCR demonstrates balanced performance by effectively suppressing hallucinations without substantially shortening the average caption length.

**AMBER.** Table 4 presents evaluation results on the AMBER (Wang et al., 2023) benchmark, which evaluates hallucinations in multimodal LLMs for both generative and discriminative visual reasoning. Specifically, DSCR reduces CHAIR$_S$ by approximately 6.35% and improves overall F1 by about 14.2% compared to the baseline. Across most category–metric combinations, DSCR consistently outperforms both the baseline and VCD, demonstrating effective hallucination mitigation and improved grounding quality.

## 3.2 QUALITATIVE RESULTS

Figure 5 visualizes selected pairs of image, depth map, and query-to-image attention map before and after applying DSCR. DSCR encourages the model to focus on more relevant regions guided by depth and spatial information. For instance, in the second image, before applying DSCR, the attention is scattered over irrelevant background regions, leading to an incorrect answer. After applying DSCR, the attention is redirected to the bicycle, resulting in the correct answer. We suggest that DSCR improves attention allocation, ensuring focus on the most relevant regions for answering the query. Additional qualitative results are provided in the Appendix.

## 4 ANALYSIS

In this section, we present a series of ablation experiments to analyze the contribution of each component in DSCR. Unless specified, experiments are conducted on the MME hallucination benchmark using LLaVA-1.5. See appendix for more ablations.

## 4.1 INFERENCE TIME AND RESOURCE USAGE

To evaluate the efficiency of the proposed method, we sample 10 images from the MME validation set and run the full DSCR pipeline (visual encoding, language decoding, cache refinement, and Depth-Anything pre-processing) five times per image to compute the average inference time. Peak GPU memory is measured across five independent runs and averaged. As shown in Table 5, DSCR achieves the fastest per-image runtime among all previous methods while maintaining GPU memory usage on par with alternatives. Furthermore, when DSCR is applied on top of existing techniques, it delivers clear performance gains with minimal extra overhead. This efficiency arises because DSCR requires only a single forward pass and one-time KV refinement, whereas several alternative methods necessitate multiple inferences for a single query (Leng et al., 2024b).

Table 5: Inference time and GPU memory consumption per image for various methods.

| Method | Time (sec/img) | GPU Mem. (MiB) |
|---|---|---|
| Baseline | 9.35 | 29950.3 |
| DSCR (Ours) | 11.06 | 32813.7 |
| VCD | 15.13 | 29979.6 |
| OPERA | 39.37 | 37717.3 |
| HALC | 31.47 | 32890.7 |
| DAMO | 11.80 | 29965.2 |
| AGLA | 23.97 | 33448.1 |
| VCD + DSCR | 16.31 | 32841.1 |
| OPERA + DSCR | 42.47 | 40569.8 |
| HALC + DSCR | 34.53 | 36704.4 |
| DAMO + DSCR | 13.48 | 32828.6 |
| AGLA + DSCR | 25.08 | 36306.0 |

Table 6: COCO Image captioning performance with and without DSCR.

| Method | BLEU-4 | CIDEr | SPICE |
|---|---|---|---|
| Baseline | 0.122 | 0.529 | 0.162 |
| **DSCR** | **0.235** | **0.909** | **0.193** |

Table 7: Hyperparameter settings used across all models and datasets.

| H.Params | $\sigma_d$ | $\sigma_s$ | $\alpha$ | $\beta$ | Layers |
|---|---|---|---|---|---|
| Value | 0.6 | 0.6 | 0.6 | 0.8 | 10–39 |

Table 8: GPU memory, per-image inference time, and performance comparisons of DSCR using different depth estimators.

| Depth Model | GPU (MiB) | Time (sec/img) | OCR | Color | Count | Existence | Position | Posters |
|---|---|---|---|---|---|---|---|---|
| Depth-Anything-v2 (Yang et al., 2024b) | 2134.1 | 1.34 | 132.5 | 175.0 | 160.0 | 195.0 | 120.0 | 140.48 |
| MiDaS-Lite (Ranftl et al., 2020) | 1264.2 | 1.15 | 125.0 | 180.0 | 160.0 | 195.0 | 111.67 | 132.65 |
| DPT-Lite (Ranftl et al., 2021) | 526.1 | 1.54 | 125.0 | 180.0 | 160.0 | 195.0 | 111.67 | 132.65 |

## 4.2 GENERALIZATION TO GENERAL VL TASKS

While many hallucination-mitigation techniques require retraining or introduce trade-offs that compromise overall task performance, DSCR is entirely training-free and even slightly improves the model's core captioning ability. On the COCO image captioning task (Table 6), DSCR shows improvements of +0.113 in BLEU-4, +0.380 in CIDEr, and +0.031 in SPICE compared to the LLaVA-1.5 baseline. These results show that DSCR effectively suppresses hallucinations while maintaining and even enhancing overall text generation quality for standard vision–language benchmarks.

## 4.3 DEPTH ESTIMATION MODELS AND HYPERPARAMETER SENSITIVITY

In Table 8, we evaluate DSCR with three monocular depth estimators (i.e., Depth-Anything v2, MiDaS-Lite, and DPT-Lite). Despite variations in GPU memory usage (526–2134 MiB) and inference time (1.15–1.54 s per image), the key metrics remained within ±7%. This minimal performance fluctuation across different depth-estimation models demonstrates that DSCR consistently mitigates hallucinations even when using noisy depth maps or an extremely lightweight model.

For all experiments using DSCR, we adopt the fixed hyperparameter settings listed in Table 7. This is justified by the observation that our method is not sensitive to hyperparameter choices, showing less than 5% performance variation across alternative settings. For detailed results on all tested hyperparameter combinations, please refer to the Appendix.

## 5 CONCLUSION

In this paper, we proposed DSCR, a novel training-free KV cache refinement method guided by depth and spatial cues. Our method mitigates hallucinations in vision-language models by reallocating KV vectors based on geometric and spatial consistency. To validate its effectiveness, we conducted comprehensive experiments on five hallucination benchmarks (MME, POPE, RePOPE, CHAIR, and AMBER), achieving up to 41.6% accuracy improvements, and can be integrated on top of existing methods. We further introduced a depth confusion mini-benchmark, specifically designed to evaluate cases where multiple objects overlap or share similar depths. Even when depth models produced inaccurate predictions, DSCR consistently improved performance by guiding attention to semantically meaningful regions. To the best of our knowledge, DSCR is the first to refine the KV cache using auxiliary geometric cues (i.e., depth and position), making it a practical and generalizable plug-in for hallucination mitigation.

## 6 ACKNOWLEDGMENT

This work was supported by Institute of Information & communications Technology Planning & Evaluation (IITP) grant funded by the Korea government(MSIT) (RS-2024-00398157) and by Institute of Information & communications Technology Planning & Evaluation (IITP) under Next-generation Cloud-native Cellular Network Leadership Program (IITP-2026-RS-2024-00418784) grant funded by the Korea government(MSIT).

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

# A PRELIMINARIES

**Internal Self-Attention Mechanism.** Self-attention (SA) module within each Transformer layer is the core component that incorporates past context to produce features for the current input. The operation of SA for each attention head can be simplified as:

$$\text{Softmax}\Big(\frac{q_t \cdot \big[K^I; K^X; K^Y\big]}{\sqrt{d_h}}\Big)\big[V^I; V^X; V^Y\big], \tag{7}$$

where $K^X$ and $K^Y$ represent the key caches corresponding to question and current answer tokens, respectively. Similarly, $V^X$ and $V^Y$ denote the associated value caches. This operation can be divided into three steps: 1) calculate the similarity between the current token's query ($q_y \in \mathbb{R}^{d_h}$) and the keys of all previous tokens, 2) apply the softmax function to the similarity scores to obtain attention probabilities, and 3) aggregate the values of all previous tokens by taking a weighted sum based on the attention probabilities. In essence, modification of $K^I$ affects the attention probabilities, while that of $V^I$ changes the output of the self-attention operation.

**Text Generation Process.** Similar to LLMs, VLMs operate in an autoregressive manner, performing the next-token prediction to generate responses. Consider a visual question-answering setup where a VLM generates an answer $Y = \{y_1, \cdots, y_T\}$ to the provided query $X = \{x_1, \cdots, x_M\}$. To generate $t$-th token $y_t$ of the answer, VLM computes the probability of the next token as:

$$P_{\text{VLM}}(y_t|y_{1:t-1}, X, I) = P_{\text{VLM}}(y_t|y_{1:t-1}, x_{1:M}, \mathbf{k}^I_{1:N}, \mathbf{v}^I_{1:N}). \tag{8}$$

This indicates that the generated text depends on the question $X$[1] and KV cache derived from the image tokens. Notably, once the KV cache is computed, users can change the query to ask different questions about the same image without recomputing the KV cache (see Figure 2 (b)). This property is referred to as query-agnostic behavior.

# B RELATED WORK

## B.1 HALLUCINATION IN VLMS

Visual hallucination in VLMs refers to the generation of content that is not grounded in the visual input, such as nonexistent objects or incorrect attributes. To mitigate this problem, various strategies have been proposed across model components, including improved pretraining data (Zhou et al., 2024), larger or more expressive vision encoders (Liu et al., 2024a; Chen et al., 2024c), and decoding-level interventions (Yu et al., 2024; Leng et al., 2024a). Among the latter, OPERA (Yu et al., 2024) penalizes overconfident attention weights during decoding to suppress ungrounded responses, while VCD (Leng et al., 2024a) filters hallucinations by comparing outputs from perturbed and original images through contrastive decoding.

## B.2 KEY-VALUE CACHE MANIPULATION

As context length increases significantly, LLM suffers from the memory bottleneck (Kim et al., 2023). In order to alleviate memory overhead and reduce inference time, various techniques have been proposed to optimize the KV cache (Bolya et al., 2023; Chen et al., 2025; Wan et al., 2024; Liu et al., 2024d; Tu et al., 2024). For example, sparsity-based cache allocation (Tu et al., 2024), attention score-based pruning (Wan et al., 2024), prioritizing recent tokens (Liu et al., 2024d), and differentiating between visual and text tokens (Tu et al., 2024) have been explored. While these methods have successfully addressed issues involved in efficiency, they have predominantly focused on compressing the cache itself rather than enhancing the representation within the cache. To our knowledge, there has been no prior work in either vision or language domains that integrates additional information into the KV cache to improve its representational strength.

---

[1]Question tokens also generate corresponding KV cache in practical systems, but we omit the details here. See LLM inference literature (Kim et al., 2023).

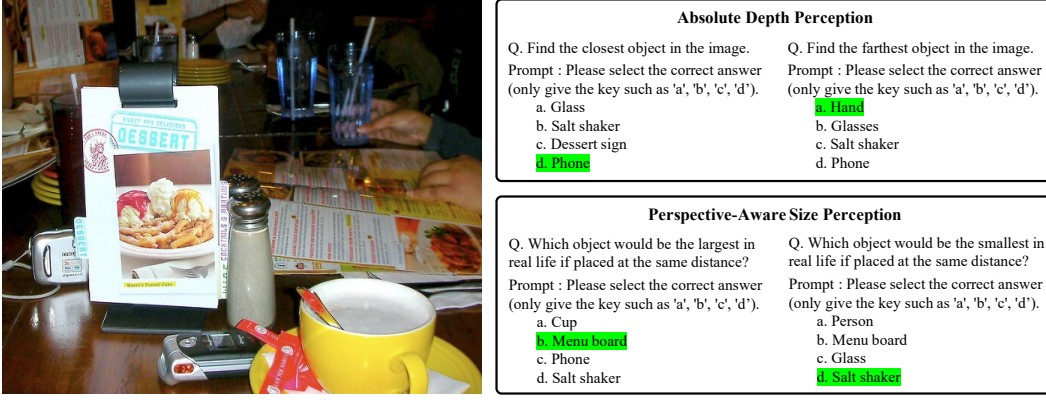

Figure 6: Example from the Depth Hallucination Mini-Benchmark . The left shows the input image, and the right displays four sample questions grouped by reasoning type. **Top**: Questions assessing *Absolute Depth Perception* (e.g., identifying the closest or farthest object in the scene). **Bottom**: Questions assessing *Perspective-Aware Size Perception*, which test whether the model can infer real-world object size based on perspective. Correct answers are highlighted in green.

### B.3 VISUAL GROUNDING IN VISION-LANGUAGE MODELS

Strengthening visual grounding is a central objective for VLMs, and a variety of approaches have been proposed toward this goal. One line of work performs spatial and geometric grounding by tying language to 3D scene structure or monocular 3D object extents. For example, SpatialVLM (Chen et al., 2024a) trains a vision–language model on large-scale synthetic spatial question and answer data generated from real images and metric 3D representations, which improves quantitative reasoning about distances, sizes, and relative positions. Mono3DVG (Zhan et al., 2023) defines 3D visual grounding in monocular RGB images and proposes a transformer-based model that jointly exploits appearance features, a dedicated depth predictor, and geometry-aware text embeddings to localize the full 3D extent of referred objects. Another line of work enhances grounding by aligning textual tokens or phrases with image regions and explicitly reusing vision tokens as evidence when scoring outputs. GroundVLP (Shen et al., 2023) fuses Grad-CAM heatmaps from a vision–language backbone with region proposals from open-vocabulary detectors to achieve zero-shot phrase grounding without task-specific grounding annotations. ReVisiT (Cho & Kim, 2025) introduces a training-free decoding strategy that projects vision tokens into the text-token distribution space, dynamically selects the most relevant vision token at each decoding step, and uses it to refine the output distribution, thereby reducing over-reliance on language priors and improving visual grounding. These methods share the common aim of tightly coupling linguistic expressions with concrete visual entities, typically through additional training or modified decoding. In contrast, DSCR does not explicitly align language with individual objects or regions and does not introduce new training or decoding stages, but instead refines internal visual key–value representations at inference time using depth and spatial priors so that hallucinations are mitigated while the backbone parameters and decoding pipeline remain unchanged.

## C EXPERIMENT SETUP

### C.1 DEPTH HALLUCINATION MINI-BENCHMARK

To evaluate the depth perception capabilities of vision-language models, we construct a new dataset called the Depth Hallucination Mini-Benchmark. The benchmark is designed to assess two key evaluation objectives. First, it tests whether the model can reason about absolute distances between objects; and second, whether it can perceive relative depth under perspective while preserving knowledge of object identity. The dataset comprises 50 images manually selected from 500 candidates in the COCO val2014 dataset (Lin et al., 2014). Images were chosen based on two criteria: they contain at least two distinct objects, and they exhibit clear perspective information, allowing for meaningful depth comparison.

For each image, we generate four depth-related questions: identifying the closest object, the farthest object, the largest object in actual size, and the smallest object in actual size. We use GPT-4o (team, 2024) to generate multiple-choice answers for each question, including one correct answer and three distractors. All generated choices are subsequently verified by humans to ensure that the objects exist in the image and that the correct answers are indeed valid.

We categorize the questions into two types based on their intended depth reasoning objective.

- **Absolute Depth Perception** includes questions about the closest and farthest objects. The purpose of this category is to evaluate whether the model can identify objects based on their actual distance from the camera.

- **Perspective-Aware Size Perception** includes questions about the largest and smallest objects in actual size. This category assesses whether the model can reason about visual scale and preserve knowledge of object identity under perspective distortion.(See Figure 6 for an example from the dataset.)

## C.2 DATASET

**MME.** The MME (Fu et al., 2023) dataset is a comprehensive benchmark designed to evaluate the performance of VLMs across various aspects including fine-grained visual cognition, visual perception, and OCR. Since our work focuses on mitigating hallucinations, we utilize subsets of MME. Specifically, we employ the *existence* and *count* subsets to evaluate object-level hallucinations, and use *position* and *color* subsets to assess attribute-level hallucinations. Each subset consists of "Yes-or-No" questions, providing a straightforward assessment of VLM's ability to recognize objects and their attributes.

**POPE & RePOPE.** The POPE (Li et al., 2023c) benchmark is specifically crafted to evaluate the hallucination in VLMs. The evaluation targets object-level visual hallucinations, particularly focusing on the existence of objects in a visual scene. The dataset consists of binary classification questions for the target object, which may or may not appear in the image. Targets are selected based on three distinct sampling settings: *random*, *popular*, and *adversarial*. In the *random* setting, non-existent objects are selected randomly. In the *popular* setting, non-existent objects are chosen from a pool of frequently appearing objects in the dataset. In the *adversarial* setting, objects that commonly co-appear but do not actually exist in an image are selected to challenge the model's perception.

RePOPE (Neuhaus & Hein, 2025) is a relabeled version of the POPE benchmark that corrects annotation errors in the MSCOCO dataset. The POPE and RePOPE datasets each comprise 500 images, with six associated questions per image. The evaluation metrics include accuracy, precision, recall, and F1-score.

**CHAIR.** The Caption Hallucination Assessment with Image Relevance (CHAIR) (Rohrbach et al., 2018) is a metric for evaluating object hallucination in image captioning. It measures the extent to which generated captions refer to objects that are not present in the ground-truth annotations of the corresponding image. CHAIR consists of two components: CHAIRs, which evaluates hallucination at the sentence level, and CHAIRi, which measures it at the instance level across multiple captions. Lower scores on both components reflect improved grounding of the caption to visual content. For evaluation, 500 images are randomly sampled from the COCO 2014 validation set. Captions are generated using various VLMs prompted with "Please describe this image in detail," with a fixed maximum token limit to ensure fair comparison.

**AMBER.** We conduct evaluation on the AMBER (Wang et al., 2023) benchmark, a recently proposed dataset designed to precisely assess hallucination in vision-language models. AMBER consists of natural image–question pairs across five task types: object existence, counting, positional reasoning, color recognition, and text reading. Each question is constructed such that the correct answer is verifiable based on the visual content alone, and any incorrect answer lacking visual grounding is regarded as a hallucination. The dataset includes a diverse range of image sources such as real-world photos, diagrams, and scene renderings, allowing for comprehensive evaluation

Table 9: Evaluation results on our Depth Hallucination Mini-Benchmark.

| Method | Abs. Depth Score | Persp.-Aware Size Score | Average Score |
|---|---|---|---|
| Baseline | 60.0 | 51.0 | 55.5 |
| VCD (Leng et al., 2024b) | 73.0 | 68.0 | 70.5 |
| OPERA (Huang et al., 2024) | 78.0 | 73.0 | 75.5 |
| **DSCR (Ours)** | **82.0** | **75.0** | **78.5** |
| VCD + Ours | 77.0 | 67.0 | 72.0 |
| OPERA + Ours | 80.0 | 72.0 | 76.0 |

across different visual domains. The primary evaluation metric is accuracy, which directly reflects a model's ability to generate visually grounded responses and avoid hallucinated outputs.

### C.3 COMPUTATIONAL RESOURCES

All experiments were conducted on a machine equipped with three NVIDIA A6000 GPUs (48GB each), an Intel Xeon Gold 6526Y processor with 32 threads (16 physical cores), and 754GB of RAM. Our method was evaluated solely in the inference setting, with no additional training or fine-tuning.

### C.4 MODELS

We employ the state-of-the-art MDE model, Depth-Anything-v2 (Yang et al., 2024b), due to its efficient inference cost and accurate depth predictions across a wide range of images. As for baselines, we use popularly used VLMs including LLaVA-1.5 (Liu et al., 2024a), LLaVA-1.6 (Liu et al., 2024b),mPLUG-Owl2 (Ye et al., 2024), Qwen-VL (Bai et al., 2023), and Qwen2.5-VL (Bai et al., 2025). All three models follow the common "vision encoder-interface-language model" framework, which utilizes a pre-trained visual encoder to extract visual tokens. All experiments were conducted with fixed hyperparameters $\sigma_d = 0.6$, $\sigma_s = 0.6$, $\alpha = 0.6$, $\beta = 0.8$, and layers 10–39.

### C.5 IMPLEMENTATION DETAILS

We reproduce both OPERA and VCD using their official implementations and verify that our re-implementation matches the original results. During reproduction, we encountered inconsistencies due to version differences in the Hugging Face Transformers library. To ensure compatibility and consistency, we standardize all experiments, including DSCR, using `Transformers` version 4.31.0. Our DSCR implementation applies cache refinement across all layers simultaneously using optimized tensor operations, which eliminate additional overhead. As a result, inference is efficient and takes approximately 1 to 3 seconds per image.

## D ADDITIONAL EXPERIMENTS

### D.1 RESULTS ON DEPTH HALLUCINATION MINI-BENCHMARK

As shown in Table 9, DSCR outperforms the baseline on the Depth Hallucination Mini-Benchmark. It improves the absolute depth perception score by 36.7% and the perspective-aware size perception score by 47.1%, resulting in a total score improvement of 41.4%. DSCR also surpasses VCD (Leng et al., 2024b) and OPERA (Huang et al., 2024) by 8.0% and 3.0%, respectively. When used as an add-on, DSCR further increases the total score of VCD by 1.5% and OPERA by 0.5%. From the experimental results, we confirm that DSCR is effective both as a standalone method and as a cascade component that enhances existing hallucination mitigation approaches.

### D.2 RESULTS ON POPE

To further evaluate the generalizability of our method, we conduct experiments on the POPE benchmark, which measures robustness to visual hallucinations across MSCOCO (Lin et al., 2014), A-OKVQA (Schwenk et al., 2022), and GQA (Hudson & Manning, 2019) datasets. We apply DSCR

Table 10: Evaluation results on the CHAIR dataset using mPLUG-Owl2 model.

| Method | CHAIR$_S$ ↓ | CHAIR$_I$ ↓ | Recall ↑ | Avg. Len. |
|---|---|---|---|---|
| Baseline | 38.6 | 11.6 | **73.0** | 88.5 |
| VCD | 45.4 | 13.6 | 68.6 | 88.9 |
| OPERA | **35.6** | 11.6 | 71.8 | 86.3 |
| **DSCR** | **35.6** | **11.2** | 69.3 | 77.0 |

Table 11: Evaluation results on the VQAv2 dataset using LLaVA-1.5 and Qwen2.5-VL.

| Metric | LLaVa-1.5 | | Qwen2.5-VL | |
|---|---|---|---|---|
| | **Baseline** | **DSCR** | **Baseline** | **DSCR** |
| Overall Accuracy | 79.87 | **80.13** | 84.27 | **84.73** |

to three vision-language models: LLaVA-1.5, Qwen-VL, and mPLUG-Owl2, under Random, Popular, and Adversarial settings.

As shown in Table 19, DSCR consistently improves all evaluation metrics, including accuracy, precision, recall, and F1-score, across different models and settings. For example, in the Random setting on the MSCOCO dataset, LLaVA-1.5 improves its F1-score from 0.82 to 0.85, Qwen-VL from 0.80 to 0.84, and mPLUG-Owl2 from 0.83 to 0.86. Similar gains are observed under the Popular and Adversarial conditions, demonstrating that DSCR robustly enhances grounding across diverse hallucination scenarios. These results confirm the effectiveness of DSCR in reducing hallucinations and improving the reliability of vision-language models across various architectures and conditions.

### D.3 COMPARISON TO OTHER METHODS

To validate the effectiveness and plug-and-play applicability of DSCR, we compare it with existing methods for mitigating visual hallucinations on the POPE benchmark. Tables 20, 21, and 22 report the results of recent approaches, including VCD, OPERA, and ours, in terms of accuracy, precision, recall, and F1-score across various settings.

Overall, DSCR consistently improves performance across all vision-language models and evaluation scenarios when applied to either VCD or OPERA. These improvements are especially notable in challenging conditions such as the Adversarial setting, where hallucination risk is elevated due to co-occurrence biases. Notably, on the MSCOCO dataset under the Random setting, applying DSCR to OPERA on mPLUG-Owl2 improves the F1-score from 0.81 to 0.87, reflecting a relative improvement of approximately 7.4%. These results highlight the effectiveness of DSCR as a general and model-agnostic refinement strategy that can be seamlessly integrated into existing methods, improving hallucination robustness without modifying model parameters.

### D.4 RESULT ON CHAIR

Table 10 presents evaluation results on the CHAIR benchmark using the mPLUG-Owl2 model. Consistent with the results in Table 3 using the LLaVA-1.5 model, DSCR achieves the lowest hallucination scores (CHAIR$_S$: 34.6, CHAIR$_I$: 10.3). Although the recall (70.3) is slightly lower than that of OPERA, DSCR effectively suppresses hallucinations while maintaining competitive recall performance. These results suggest that the effectiveness of DSCR in reducing hallucinations generalizes across different backbone models.

### D.5 RESULT ON VQAv2

To investigate the effect of DSCR on general VL tasks, we conduct additional experiments on question-answering task.ca Table 11 shows evaluation results on the VQAv2 dataset. We randomly sample 500 examples from the VQAv2 validation set and evaluate the overall accuracy of baseline models and DSCR on this subset. In both LLaVa-1.5 and Qwen-2.5-VL, DSCR preserves the

Table 12: Ablation study for varying the size of depth estimation model, conducted on the MME dataset using LLaVA-1.5 model.

| Model | Type (Size) | Object-level | | Attribute-level | | Total |
|---|---|---|---|---|---|---|
| | | Existence | Count | Position | Color | |
| - | - | 173.33 | 116.66 | 113.33 | 123.33 | 526.66 |
| Depth-Anything-v2 | Small (0.03B) | 190.00 | 153.33 | 120.00 | 170.00 | 633.33 |
| Depth-Anything-v2 | Base (0.1B) | 190.00 | 160.00 | 120.00 | 170.00 | 640.00 |
| Depth-Anything-v2 | Large (0.3B) | **195.00** | **160.00** | **120.00** | **175.00** | **650.00** |

Table 13: Ablation study on the effect of $\sigma$, the size of the Gaussian kernel.

| Sigma | Object-level | | Attribute-level | | Total |
|---|---|---|---|---|---|
| | Existence | Count | Position | Color | |
| 0.2 | 190.00 | 153.33 | 120.00 | 170.00 | 633.33 |
| 0.4 | 190.00 | 155.00 | 120.00 | 170.00 | 635.00 |
| **0.6** | **195.00** | **160.00** | **120.00** | **175.00** | **650.00** |
| 0.8 | 190.00 | 155.00 | 120.00 | 170.00 | 635.00 |

Table 14: Ablation study on the selection of Key and Value Cache.

| Setting | Object-level | | Attribute-level | | Total |
|---|---|---|---|---|---|
| | Existence | Count | Position | Color | |
| Value-only | 190.00 | **160.00** | **120.00** | 170.00 | 640.00 |
| **Key-Value** | **195.00** | **160.00** | **120.00** | **175.00** | **650.00** |
| **Key-only** | **195.00** | **160.00** | **120.00** | **175.00** | **650.00** |

Table 15: Ablation study on the combination of external sources.

| Setup | Object-level | | Attribute-level | | Total |
|---|---|---|---|---|---|
| | Existence | Count | Position | Color | |
| Depth (D) | 195.00 | 160.00 | 120.00 | 170.00 | 645.00 |
| Spatial (S) | 195.00 | 160.00 | 120.00 | 170.00 | 645.00 |
| **D & S** | **195.00** | **160.00** | **120.00** | **175.00** | **650.00** |

Table 16: Ablation study on the position of layers to apply DSCR.

| Layers | Object-level | | Attribute-level | | Total |
|---|---|---|---|---|---|
| | Existence | Count | Position | Color | |
| 10–20 | 190.00 | 158.33 | 120.00 | 170.00 | 638.33 |
| 10–30 | 195.00 | 160.00 | 120.00 | 170.00 | 645.00 |
| 20–30 | 190.00 | 153.33 | 120.00 | 170.00 | 633.33 |
| 20–40 | 190.00 | 160.00 | 120.00 | 170.00 | 640.00 |
| **10–39** | **195.00** | **160.00** | **120.00** | **175.00** | **650.00** |

overall accuracy compared to the baseline models. These results demonstrate that DSCR mitigates hallucination without degrading the models' general VL capabilities.

# E ABLATION STUDIES

## E.1 DEPTH MODEL SIZE

We conduct an ablation study to analyze the effect of the quality of estimated depth by varying the size of the monocular depth estimation model. Table 12 presents the results for different versions of the Depth-Anything-v2 (Yang et al., 2024b) model, ranging from Small (0.03B) to Large (0.3B) size models. The results indicate that depth information has a considerable impact on reducing hallucinations in VLMs. Notably, even the smallest depth model provides a substantial performance boost, demonstrating depth information's critical role in enhancing visual representations.

### E.1.1 DEPTH VS. SPATIAL PROXIMITY

Table 15 reports results for depth-only, spatial-only, and combined depth&spatial weighting. The combined "D&S" setting yields the best total score of 650.00, confirming that depth cues and 2D proximity complement each other.

## E.2 HYPERPARAMETER SETTING

We conduct ablation studies focusing on four key aspects: the scale of the Gaussian function for determining adjustment sensitivity, the strategy for cache refinement (modifying the Key, Value, or

Table 17: The architecture comparison for the VLMs and the MDE model used in the experiments.

| Model | Vision Encoder | | | LLM | |
|---|---|---|---|---|---|
| | Type | Size | Input | Type | Size |
| LLaVA-1.5 (Liu et al., 2024c) | VIT-L/14 | 0.3B | 336x336 | Vicuna | 7B |
| Qwen-VL (Bai et al., 2023) | VIT-G/14 | 1.9B | 336x336 | Qwen | 7B |
| mPLUG-Owl2 (Ye et al., 2024) | VIT-L/14 | 0.3B | 336x336 | LLaMA | 7B |
| Depth-Anything-v2 (Yang et al., 2024b) | VIT-L/14 | 0.3B | 518x518 | - | - |

both), and the selection of Transformer layer positions for modification. All subsequent experiments utilize LLaVA-1.5 on the MME hallucination benchmark.

### E.3 GAUSSIAN KERNEL SIZE

We first evaluate the effect of the Gaussian scale parameters $\sigma_d$ and $\sigma_s$ from Eq. (1) and Eq. (2), setting $\sigma_d = \sigma_s = \sigma$ for simplicity. Here, $\sigma$ controls how broadly neighboring tokens influence each other in the cache refinement. As shown in Table 13, $\sigma=0.6$ provides the best balance between local detail sensitivity and cache smoothness.

### E.4 KEY-VALUE REFINEMENT STRATEGY

Next, we compare three strategies: Value-only, Key-only, and Key+Value, as shown in Table 14. While all three settings improve performance, Key-only refinement achieves the highest score, slightly outperforming the Key+Value variant. This implies that key vectors play a more critical role in mitigating hallucinations, and that updating only the Key can provide an efficient trade-off between performance and computational cost.

### E.5 REFINEMENT LAYER RANGE

Finally, we investigate which layers benefit most from cache refinement. As shown in Table 16, applying KV cache refinement to layers 10–39 provides the best performance, suggesting that mid-to-high Transformer layers are most effective targets for reducing hallucinations.

### E.6 QUALITATIVE ANALYSIS OF FAILURE AND SUCCESS CASES

To better understand the behavior of DSCR under challenging scenarios, we present qualitative analyses of three samples using various depth estimators (Depth-Anything v2, DPT-Lite, MiDaS-Lite), as shown in Figure 7.

In the first case, all models fail to predict the correct answer, including DSCR. Although the depth-based refinement encourages attention toward text regions, the baseline model lacks sufficient language capability to correctly interpret the text content. This highlights a limitation of DSCR—it can guide attention to semantically relevant regions, but cannot compensate for fundamental weaknesses in the underlying language reasoning.

In contrast, the second and third examples show how DSCR successfully mitigates hallucinations. Even when lightweight depth models like DPT-Lite and MiDaS-Lite generate noisy or low-quality depth maps, DSCR still improves prediction by redirecting focus to plausible object regions. When more accurate depth models (e.g., Depth-Anything v2) are used, performance further improves.

Importantly, DSCR remains compatible with lightweight depth estimators, enabling deployment in real-world applications where compute resources are limited. Moreover, in practical scenarios involving depth cameras (e.g., AR glasses or mobile devices), depth can be directly obtained without requiring an additional prediction model—potentially leading to even more reliable results.

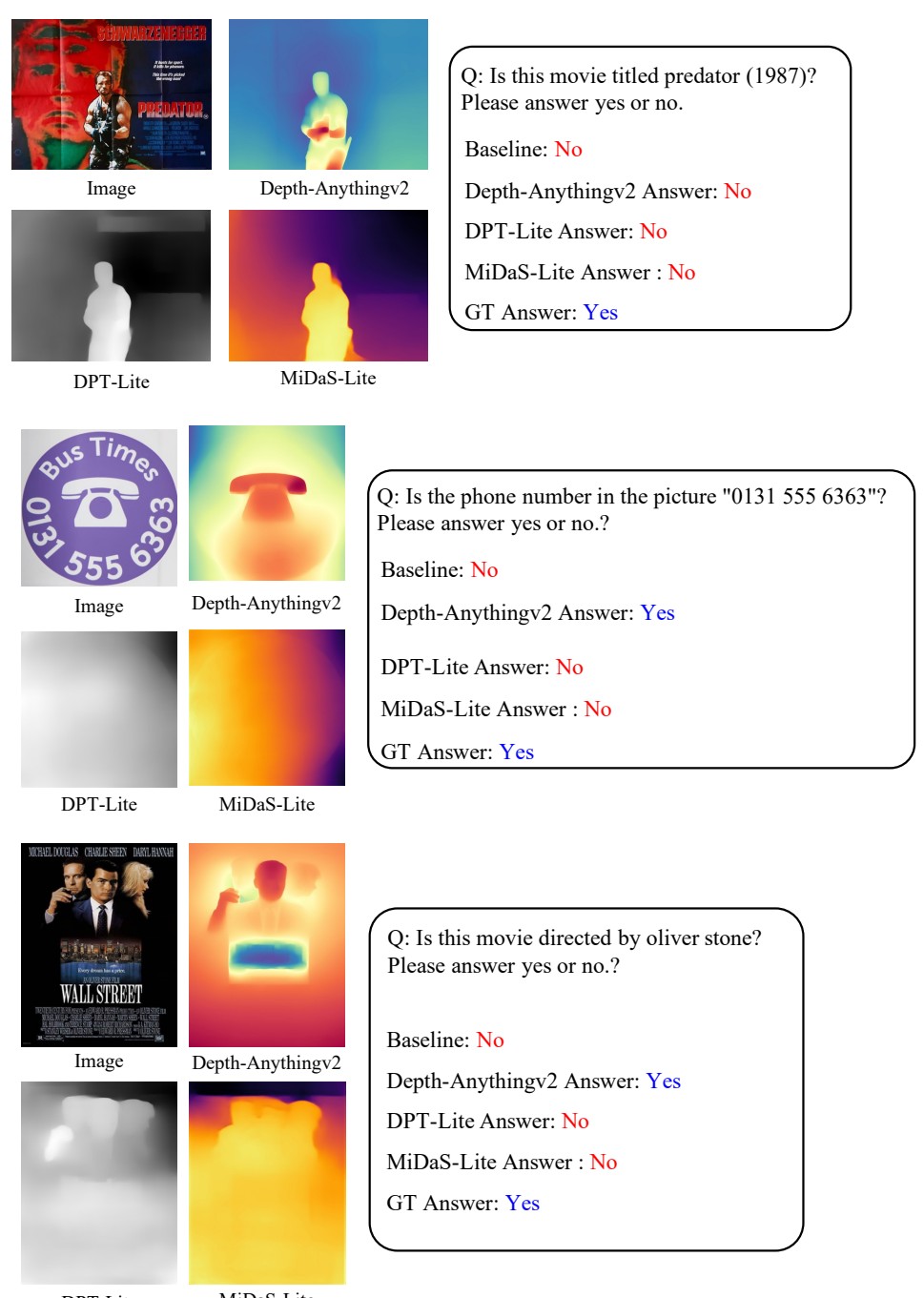

Figure 7: Qualitative examples show the effect of DSCR across different depth estimation conditions. (Top) All models, including DSCR, fail due to the baseline's inability to interpret the text despite depth-guided attention. (Middle, Bottom) DSCR correctly answers the question by focusing on relevant regions, even when using noisy depth maps from lightweight models like DPT-Lite or MiDaS-Lite. Performance further improves with more accurate depths (e.g., Depth-Anything v2).

### E.7 ADDITIONAL COST OF DEPTH ESTIMATION

We show that the additional cost of DSCR is minimal. As detailed in Table 17, Depth-Anything-v2 model employed for depth estimation contains 0.3B parameters, comparable to the vision encoders and significantly smaller than LLMs used in VLMs. Additionally, the depth estimation process

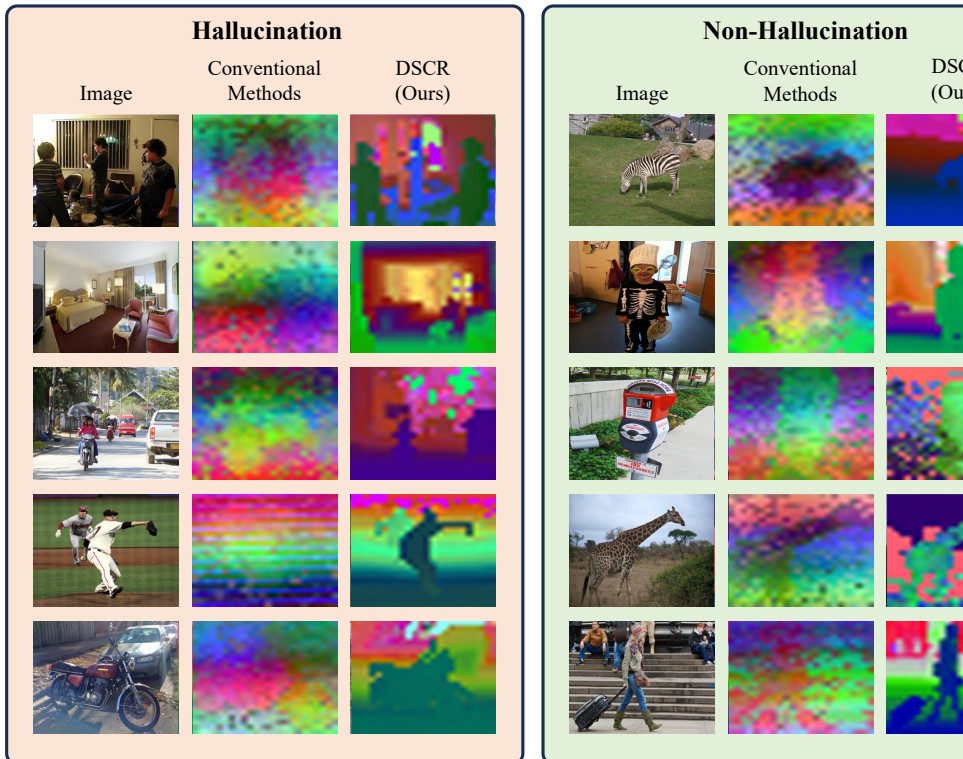

Figure 8: Key vector visualizations for hallucinated (left) and non-hallucinated (right) cases. Each group shows the input image, key vector visualization from conventional methods, and key vector visualization from DSCR (ours). In hallucinated cases, conventional methods produce disorganized key vectors, while DSCR yields more semantically aligned representations. In non-hallucinated examples, both methods generate coherent vectors, but DSCR maintains sharper object boundaries and spatial structure.

with the MDE model is a one-time operation per image, requiring only a few tens of milliseconds on a GPU (Yang et al., 2024a). This implies that integrating DSCR into existing VLM frameworks introduces negligible computational overhead, thereby maintaining the overall system's performance and scalability.

### E.8 MORE QUALITATIVE RESULTS

Figure 8 extends the key vector visualizations shown in Figure 1 to a broader set of examples. The left half shows hallucinated cases, while the right half displays non-hallucinated ones. Each row presents the input image, the key vector visualization from conventional methods, and the corresponding visualization after applying DSCR. In hallucinated cases, conventional methods exhibit disorganized and noisy vector patterns, whereas DSCR produces more structured representations with clearer object boundaries and spatial organization. These results qualitatively support the effectiveness of DSCR in enhancing visual coherence and improving object-centric reasoning.

Figure 9 visualizes additional examples. Results show that DSCR successfully identifies the important objects and encourages attention mechanism to focus more on the relevant regions. For example, in the first and last rows of the Figure 9, we can observe that object (dog, bicycle) region stands out after applying DSCR; in contrast, the original model pays more attention to background and unrelated image patches.

We empirically observe that attention is biased towards first several image tokens, represented as highlighted upper regions in the attention heatmap. We assume that this is partially because LLM processes image tokens sequentially. Please note that similar phenomenon, termed Attention Sink, has been also discovered in text-only LLMs (Xiao et al., 2024).

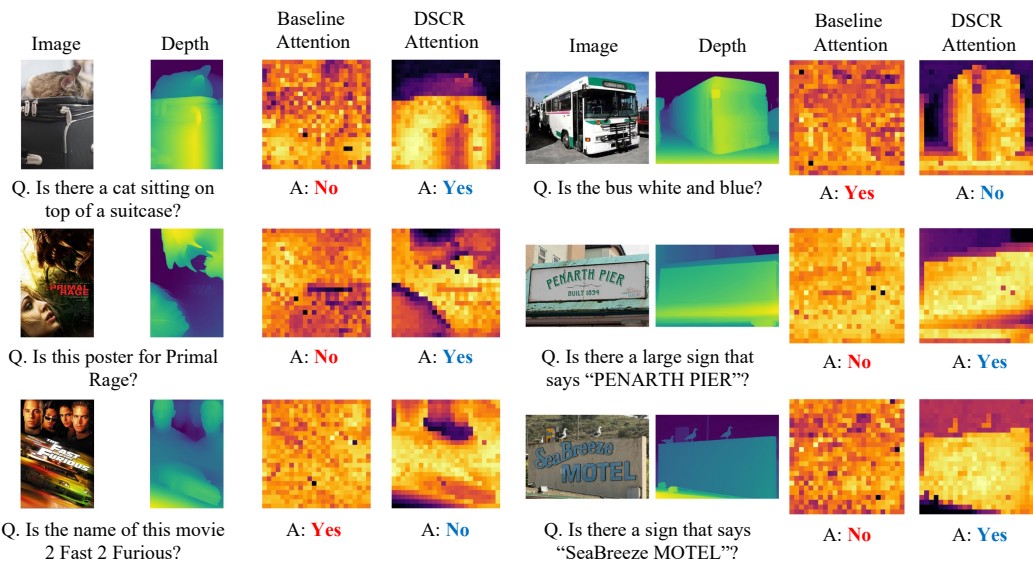

Figure 9: VQA examples, including image, depth, and query-to-image attention heatmaps before and after applying DSCR.

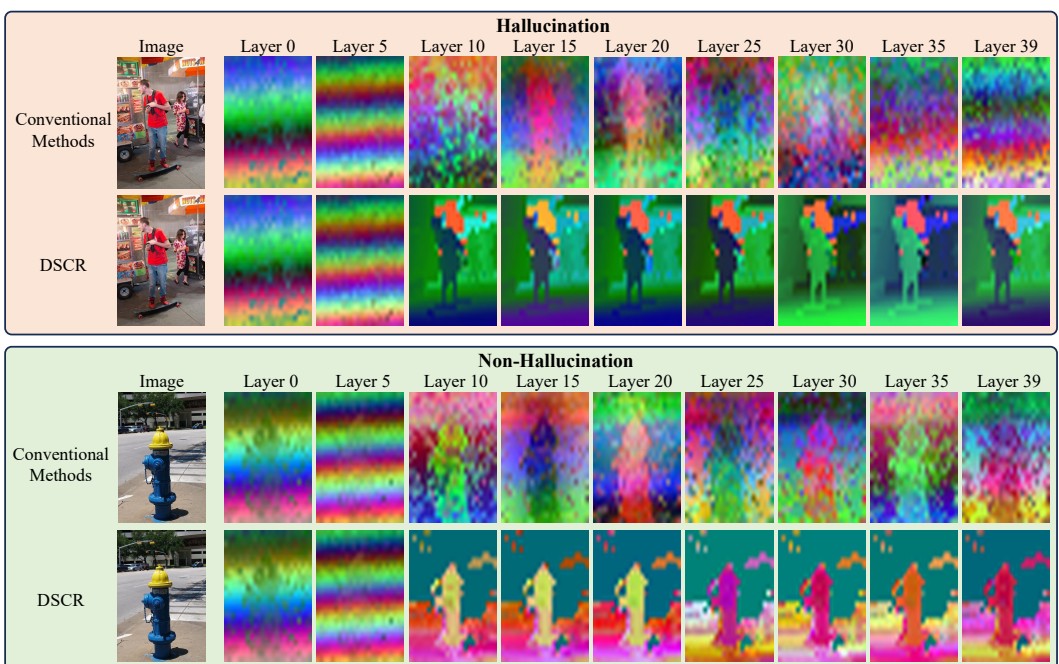

Figure 10: PCA visualizations of key vectors across layers for a hallucination example (top) and a non-hallucination example (bottom). In each block, the top row is the baseline model and the bottom row is DSCR. The visualization shows that DSCR produces more object-aligned patterns and clearer separation between foreground and background, especially in middle and upper layers.

Figure 10 shows how DSCR changes the key vectors across layers. For hallucination-occurred and non-hallucinated examples, we project visual key vectors at layers 0, 5, 10, 15, 20, 25, 30, 35, and 39 of the baseline model and from DSCR. Since DSCR is applied to layers 10–39, the visualizations at layers 0 and 5 are identical in both rows. From layer 10 onward, the baseline shows stripe-like or speckled patterns without a clear object shape, especially in the hallucination case. In contrast, DSCR shows interpretable patterns where object contours and the background are clearly separated

Table 18: Boundary Contrast (BC) statistics on the MME hallucination subsets and the Depth Hallucination Mini-Benchmark. For the LLaVA-1.5 model, DSCR increases $\text{BC}_{\text{in}}$ and decreases $\text{BC}_{\text{out}}$ on hallucination-fix cases (i.e., corrected by DSCR), which leads to a larger overall BC difference compared to the baseline. There is no cases such that DSCR reverted the correct answer.

| Split | Method | $\text{BC}_{\text{in}}$ | $\text{BC}_{\text{out}}$ | BC |
|---|---|---|---|---|
| Baseline wrong, DSCR correct | Baseline | 0.71 | 0.64 | 0.07 |
| | DSCR | 0.87 | 0.57 | 0.30 |
| Both wrong | Baseline | 0.69 | 0.63 | 0.06 |
| | DSCR | 0.73 | 0.60 | 0.13 |
| Both correct | Baseline | 0.82 | 0.61 | 0.21 |
| | DSCR | 0.88 | 0.60 | 0.28 |
| Baseline correct, DSCR wrong | Baseline | n/a | n/a | n/a |
| | DSCR | n/a | n/a | n/a |

across multiple layers. In the non-hallucination case, the baseline already reveals a rough object structure. DSCR further sharpens the object silhouette and suppresses background variation. These layer-wise visualizations support our claim that DSCR keeps early visual representations unchanged and restores a coherent, object-centered key structure in the mid-to-deep layers where it is applied.

### E.9    RELATIONSHIP TO CONDITIONAL RANDOM FIELD

The proposed DSCR method shares similarities with the Fully-Connected Conditional Random Field (FC-CRF) algorithm, widely used in vision applications such as image segmentation (Zheng et al., 2015; Chen et al., 2017). In short, FC-CRF also utilizes Gaussian functions to convert distances into proximity scores. However, while FC-CRF utilizes label prediction probabilities and pixel intensity for proximity computation, DSCR leverages depth information to guide cache refinement. Moreover, FC-CRF requires multiple iterations and heavy computations to achieve optimal results without changing the model's internal representation. In contrast, DSCR accomplishes effective cache refinement with a single computation step across all target layers.

### E.10    BOUNDARY CONTRAST ANALYSIS

To quantitatively investigate whether DSCR reflects object boundaries in the key vector space, we introduce a depth-defined **Boundary Contrast (BC)** metric. This metric compensates the previous PCA-based qualitative evaluation. In essence, we leverage depth maps as a proxy for identifying object boundaries in the image plane. Large depth difference between neighboring patches implies the existence of object boundary. We then measure whether key vectors along these proxy boundaries exhibit stronger contrast between same-object neighbors and different-object neighbors.

**Depth-based boundary tokens.**    For each image, we first resize the depth map to the $24 \times 24$ grid that corresponds to the visual tokens used by the VLM. Let $p$ index a grid location and let $d_p$ denote the depth value at $p$. We define the four-connected neighborhood of $p$ as

$$N(p) = \{q \mid q \text{ is the up, down, left, or right neighbor of } p\}. \tag{9}$$

We then compute a simple depth gradient magnitude

$$G_p = \max_{q \in N(p)} |d_p - d_q|. \tag{10}$$

Tokens with large $G_p$ are likely to lie on object boundaries. We define the boundary token set $B$ by thresholding $G_p$ with a percentile on the image-specific distribution

$$B = \{p \mid G_p \geq \tau_{\text{boundary}}\}, \tag{11}$$

where $\tau_{\text{boundary}}$ is chosen as, for example, the 90-th percentile of $\{G_p\}_p$ in each image. This percentile scheme adapts to the dynamic range of each depth map and yields a consistent number of boundary tokens across images.

**Inner and outer neighbors on the boundary.** For each boundary token $p \in B$, we further split its local neighbors into inner and outer sets based on depth similarity. We use per-image thresholds $\tau_{\text{in}}$ and $\tau_{\text{out}}$ that are also defined as percentiles of the distribution of $|d_p - d_q|$ over all $(p, q)$ pairs for $q \in N(p)$. Concretely,

$$N_{\text{in}}(p) = \{q \in N(p) \mid |d_p - d_q| \leq \tau_{\text{in}}\}, \tag{12}$$

$$N_{\text{out}}(p) = \{q \in N(p) \mid |d_p - d_q| \geq \tau_{\text{out}}\}. \tag{13}$$

Intuitively, $N_{\text{in}}(p)$ contains neighbors at similar depth that are likely to belong to the same object, while $N_{\text{out}}(p)$ contains neighbors with large depth jumps that are likely to cross an object boundary. We only keep boundary tokens for which both sets are non-empty.

**Boundary Contrast of key vectors.** Let $k_p^{(\ell)}$ denote the key vector at location $p$ in layer $\ell$, after averaging over attention heads and $\ell_2$-normalizing the vector. For a boundary token $p$ in layer $\ell$, we define its Boundary Contrast as

$$\text{BC}^{(\ell)}(p) = \frac{1}{|N_{\text{in}}(p)|} \sum_{q \in N_{\text{in}}(p)} \cos\big(k_p^{(\ell)}, k_q^{(\ell)}\big) - \frac{1}{|N_{\text{out}}(p)|} \sum_{q \in N_{\text{out}}(p)} \cos\big(k_p^{(\ell)}, k_q^{(\ell)}\big). \tag{14}$$

The first term measures how strongly the key vector at $p$ aligns with depth-similar neighbors inside the same object. The second term measures its vector similarity to depth-distant neighbors that are likely to belong to other objects.

For each layer $\ell$, we aggregate the BC values over boundary tokens:

$$\text{BC}^{(\ell)} = \frac{1}{|B|} \sum_{p \in B} \text{BC}^{(\ell)}(p), \quad \text{BC}_{\text{in}}^{(\ell)} = \frac{1}{|B|} \sum_{p \in B} \frac{1}{|N_{\text{in}}(p)|} \sum_{q \in N_{\text{in}}(p)} \cos\big(k_p^{(\ell)}, k_q^{(\ell)}\big), \tag{15}$$

$$\text{BC}_{\text{out}}^{(\ell)} = \frac{1}{|B|} \sum_{p \in B} \frac{1}{|N_{\text{out}}(p)|} \sum_{q \in N_{\text{out}}(p)} \cos\big(k_p^{(\ell)}, k_q^{(\ell)}\big), \quad \text{BC}^{(\ell)} = \text{BC}_{\text{in}}^{(\ell)} - \text{BC}_{\text{out}}^{(\ell)}. \tag{16}$$

Finally, we report image-level BC by averaging over a set of layers $\mathcal{L}$ where DSCR is applied:

$$\text{BC}_{\text{in}} = \frac{1}{|\mathcal{L}|} \sum_{\ell \in \mathcal{L}} \text{BC}_{\text{in}}^{(\ell)}, \quad \text{BC}_{\text{out}} = \frac{1}{|\mathcal{L}|} \sum_{\ell \in \mathcal{L}} \text{BC}_{\text{out}}^{(\ell)}, \quad \text{BC} = \frac{1}{|\mathcal{L}|} \sum_{\ell \in \mathcal{L}} \text{BC}^{(\ell)}. \tag{17}$$

**Setup and findings.** We compute $(\text{BC}_{\text{in}}, \text{BC}_{\text{out}}, \text{BC})$ for both the baseline VLM and its DSCR-augmented variant. Following the main experiments, we focus on the MME hallucination subsets and further split examples into four groups according to answer correctness: baseline wrong vs. DSCR correct, both wrong, both correct, and baseline correct vs. DSCR wrong. Table 18 summarizes the results.

Across the hallucination-fix group where the baseline hallucinates but DSCR produces the correct answer, DSCR consistently increases $\text{BC}_{\text{in}}$ and decreases $\text{BC}_{\text{out}}$, which leads to a larger overall BC compared to the baseline. In contrast, for examples where both models already answer correctly, BC remains nearly unchanged. Taken together, these trends show that DSCR specifically sharpens key vector boundaries along depth-defined object contours on challenging hallucination cases, rather than globally distorting the internal representation geometry. This quantitative evidence complements our PCA visualizations and directly supports the mechanism that DSCR restores local object consistency in the key space.

# F LIMITATIONS AND FUTURE WORKS

Despite its advantages, DSCR presents certain limitations that offer avenues for future research. Currently, DSCR is primarily applicable to VLM architectures similar to LLaVA (Liu et al., 2024c), and its effectiveness with other models, such as BLIP-like VLMs (Li et al., 2023a) that utilize Q-formers, remains unexplored. Extending DSCR to these architectures could significantly broaden its applicability. Additionally, while DSCR operates in a query-agnostic manner, incorporating query-aware processing or adopting few-shot in-context learning techniques may further enhance its performance and adaptability to diverse tasks. Furthermore, although DSCR employs a lightweight depth estimation model, integrating it more tightly with VLMs through joint training during instruction tuning could potentially improve the overall performance.

Table 19: Evaluation results of POPE benchmark.

| Dataset | Setting | Model | w/DSCR | Accuracy | Precision | Recall | F1-Score |
|---|---|---|---|---|---|---|---|
| **MSCOCO** (Lin et al., 2014) | **Random** | LLaVA-1.5 | × | **0.88** | **0.97** | **0.79** | **0.87** |
| | | | ✓ | 0.87 | **0.97** | **0.79** | 0.86 |
| | | Qwen-VL | × | **0.87** | 0.94 | **0.80** | **0.86** |
| | | | ✓ | **0.87** | **0.98** | 0.76 | **0.86** |
| | | mPLUG-Owl2 | × | **0.86** | 0.95 | **0.76** | **0.85** |
| | | | ✓ | 0.85 | **0.96** | 0.74 | 0.83 |
| | **Popular** | LLaVA-1.5 | × | **0.87** | **0.94** | **0.79** | **0.86** |
| | | | ✓ | 0.86 | **0.94** | 0.77 | 0.85 |
| | | Qwen-VL | × | **0.87** | **0.94** | **0.80** | **0.86** |
| | | | ✓ | 0.86 | **0.94** | 0.76 | 0.85 |
| | | mPLUG-Owl2 | × | **0.84** | 0.91 | **0.76** | **0.83** |
| | | | ✓ | **0.84** | **0.93** | 0.74 | 0.82 |
| | **Adversarial** | LLaVA-1.5 | × | **0.84** | **0.89** | **0.79** | **0.84** |
| | | | ✓ | **0.84** | **0.89** | 0.77 | 0.83 |
| | | Qwen-VL | × | **0.84** | 0.88 | **0.80** | **0.84** |
| | | | ✓ | **0.84** | **0.91** | 0.76 | 0.83 |
| | | mPLUG-Owl2 | × | **0.82** | 0.86 | **0.76** | **0.81** |
| | | | ✓ | **0.82** | **0.88** | 0.73 | 0.80 |
| **A-OKVQA** (Schwenk et al., 2022) | **Random** | LLaVA-1.5 | × | 0.79 | 0.73 | 0.91 | 0.81 |
| | | | ✓ | **0.85** | **0.78** | **0.96** | **0.86** |
| | | Qwen-VL | × | 0.86 | 0.94 | 0.76 | 0.84 |
| | | | ✓ | **0.87** | **0.96** | **0.77** | **0.85** |
| | | mPLUG-Owl2 | × | 0.80 | 0.74 | 0.90 | 0.82 |
| | | | ✓ | **0.85** | **0.82** | **0.91** | **0.86** |
| | **Popular** | LLaVA-1.5 | × | 0.74 | 0.68 | 0.91 | 0.78 |
| | | | ✓ | **0.79** | **0.72** | **0.96** | **0.82** |
| | | Qwen-VL | × | 0.85 | 0.93 | **0.76** | 0.83 |
| | | | ✓ | **0.86** | **0.95** | **0.76** | **0.84** |
| | | mPLUG-Owl2 | × | 0.72 | 0.67 | 0.90 | 0.76 |
| | | | ✓ | **0.81** | **0.76** | **0.91** | **0.82** |
| | **Adversarial** | LLaVA-1.5 | × | 0.67 | 0.62 | 0.91 | 0.74 |
| | | | ✓ | **0.69** | **0.63** | **0.96** | **0.76** |
| | | Qwen-VL | × | 0.80 | 0.82 | 0.76 | 0.79 |
| | | | ✓ | **0.81** | **0.84** | **0.77** | **0.80** |
| | | mPLUG-Owl2 | × | 0.68 | 0.62 | 0.90 | 0.74 |
| | | | ✓ | **0.71** | **0.66** | **0.91** | **0.76** |
| **GQA** (Hudson & Manning, 2019) | **Random** | LLaVA-1.5 | × | **0.90** | 0.92 | **0.87** | **0.89** |
| | | | ✓ | **0.90** | **0.93** | 0.85 | **0.89** |
| | | Qwen-VL | × | 0.84 | 0.84 | **0.83** | **0.84** |
| | | | ✓ | **0.85** | **0.91** | 0.77 | **0.84** |
| | | mPLUG-Owl2 | × | **0.85** | 0.91 | **0.77** | **0.84** |
| | | | ✓ | 0.84 | **0.93** | 0.75 | 0.83 |
| | **Popular** | LLaVA-1.5 | × | 0.86 | 0.84 | **0.87** | **0.86** |
| | | | ✓ | **0.87** | **0.87** | 0.85 | **0.86** |
| | | Qwen-VL | × | 0.72 | 0.68 | **0.83** | 0.75 |
| | | | ✓ | **0.80** | **0.81** | 0.77 | **0.79** |
| | | mPLUG-Owl2 | × | **0.79** | 0.80 | **0.77** | **0.79** |
| | | | ✓ | **0.79** | **0.82** | 0.75 | 0.78 |
| | **Adversarial** | LLaVA-1.5 | × | 0.81 | 0.78 | **0.87** | **0.82** |
| | | | ✓ | **0.82** | **0.79** | 0.85 | **0.82** |
| | | Qwen-VL | × | 0.75 | 0.72 | **0.77** | 0.77 |
| | | | ✓ | **0.78** | **0.79** | **0.77** | **0.78** |
| | | mPLUG-Owl2 | × | **0.77** | 0.76 | **0.77** | **0.77** |
| | | | ✓ | **0.77** | **0.79** | 0.75 | **0.77** |

Table 20: Evaluation results of POPE on MSCOCO across different settings.

| Setting | Model | Method | w/DSCR | Accuracy | Precision | Recall | F1-Score |
|---|---|---|---|---|---|---|---|
| **Random** | LLaVA-1.5 | VCD | × | **0.89** | **0.93** | **0.86** | **0.89** |
| | | | ✓ | **0.89** | **0.93** | 0.84 | **0.89** |
| | | OPERA | × | **0.88** | **0.97** | **0.79** | **0.87** |
| | | | ✓ | 0.87 | **0.97** | 0.76 | 0.85 |
| | Qwen-VL | VCD | × | **0.89** | **0.93** | **0.84** | **0.88** |
| | | | ✓ | 0.88 | **0.93** | 0.83 | **0.88** |
| | | OPERA | × | **0.88** | **0.98** | **0.84** | **0.82** |
| | | | ✓ | **0.88** | **0.98** | 0.83 | 0.81 |
| | mPLUG-Owl2 | VCD | × | **0.86** | 0.95 | **0.76** | **0.85** |
| | | | ✓ | 0.85 | **0.96** | 0.74 | 0.83 |
| | | OPERA | × | **0.86** | 0.95 | **0.75** | **0.83** |
| | | | ✓ | 0.85 | **0.96** | 0.74 | **0.83** |
| **Popular** | LLaVA-1.5 | VCD | × | **0.87** | **0.94** | **0.79** | **0.86** |
| | | | ✓ | 0.86 | **0.94** | 0.77 | 0.85 |
| | | OPERA | × | **0.87** | **0.94** | **0.79** | **0.86** |
| | | | ✓ | 0.86 | **0.94** | 0.76 | 0.84 |
| | Qwen-VL | VCD | × | **0.88** | **0.91** | **0.84** | **0.87** |
| | | | ✓ | 0.87 | **0.91** | 0.83 | **0.87** |
| | | OPERA | × | **0.87** | 0.91 | **0.84** | **0.86** |
| | | | ✓ | **0.87** | **0.92** | 0.82 | **0.86** |
| | mPLUG-Owl2 | VCD | × | **0.85** | 0.91 | **0.76** | **0.83** |
| | | | ✓ | 0.84 | **0.94** | 0.72 | 0.82 |
| | | OPERA | × | **0.84** | 0.90 | **0.76** | **0.82** |
| | | | ✓ | **0.84** | **0.92** | 0.73 | **0.82** |
| **Adversarial** | LLaVA-1.5 | VCD | × | **0.83** | 0.81 | **0.85** | **0.83** |
| | | | ✓ | **0.83** | **0.83** | 0.84 | **0.83** |
| | | OPERA | × | **0.84** | **0.89** | **0.79** | **0.84** |
| | | | ✓ | **0.84** | **0.89** | 0.76 | **0.84** |
| | Qwen-VL | VCD | × | 0.84 | 0.85 | **0.84** | **0.84** |
| | | | ✓ | **0.85** | **0.86** | 0.83 | **0.84** |
| | | OPERA | × | 0.84 | 0.84 | **0.84** | **0.84** |
| | | | ✓ | **0.85** | **0.85** | 0.83 | **0.84** |
| | mPLUG-Owl2 | VCD | × | **0.82** | 0.86 | **0.76** | **0.81** |
| | | | ✓ | **0.82** | **0.88** | 0.73 | 0.80 |
| | | OPERA | × | **0.82** | 0.87 | **0.75** | **0.80** |
| | | | ✓ | **0.82** | **0.88** | 0.74 | **0.80** |

Table 21: Evaluation results of POPE on A-OKVQA across different settings.

| Setting | Model | Method | w/DSCR | Accuracy | Precision | Recall | F1-Score |
|---|---|---|---|---|---|---|---|
| **Random** | LLaVA-1.5 | VCD | × | 0.87 | 0.87 | 0.87 | 0.87 |
| | | | ✓ | **0.87** | **0.88** | **0.86** | **0.87** |
| | | OPERA | × | 0.88 | 0.86 | 0.92 | 0.89 |
| | | | ✓ | **0.90** | **0.89** | **0.96** | **0.91** |
| | Qwen-VL | VCD | × | 0.89 | 0.88 | 0.90 | 0.89 |
| | | | ✓ | **0.88** | **0.89** | **0.88** | **0.88** |
| | | OPERA | × | 0.86 | 0.95 | 0.77 | 0.85 |
| | | | ✓ | **0.87** | **0.96** | **0.78** | **0.86** |
| | mPLUG-Owl2 | VCD | × | 0.86 | 0.91 | 0.81 | 0.85 |
| | | | ✓ | **0.86** | **0.94** | 0.76 | **0.84** |
| | | OPERA | × | 0.86 | 0.85 | 0.88 | 0.86 |
| | | | ✓ | **0.87** | **0.86** | **0.89** | **0.87** |
| **Popular** | LLaVA-1.5 | VCD | × | 0.84 | 0.83 | 0.87 | 0.85 |
| | | | ✓ | **0.84** | **0.83** | **0.86** | **0.85** |
| | | OPERA | × | 0.83 | 0.78 | 0.92 | 0.84 |
| | | | ✓ | **0.86** | **0.82** | **0.96** | **0.87** |
| | Qwen-VL | VCD | × | 0.87 | 0.86 | 0.89 | 0.88 |
| | | | ✓ | **0.87** | **0.87** | **0.88** | **0.87** |
| | | OPERA | × | 0.86 | 0.94 | 0.77 | 0.84 |
| | | | ✓ | **0.87** | **0.95** | **0.78** | **0.85** |
| | mPLUG-Owl2 | VCD | × | 0.83 | 0.84 | 0.81 | 0.83 |
| | | | ✓ | **0.82** | **0.88** | 0.76 | **0.81** |
| | | OPERA | × | 0.81 | 0.77 | 0.88 | 0.82 |
| | | | ✓ | **0.82** | **0.79** | **0.89** | **0.83** |
| **Adversarial** | LLaVA-1.5 | VCD | × | 0.78 | 0.73 | 0.87 | 0.80 |
| | | | ✓ | **0.78** | **0.74** | **0.86** | **0.80** |
| | | OPERA | × | 0.74 | 0.68 | 0.92 | 0.78 |
| | | | ✓ | **0.73** | **0.72** | **0.90** | **0.78** |
| | Qwen-VL | VCD | × | 0.81 | 0.76 | 0.89 | 0.82 |
| | | | ✓ | **0.81** | **0.77** | **0.88** | **0.82** |
| | | OPERA | × | 0.81 | 0.83 | 0.77 | 0.80 |
| | | | ✓ | **0.82** | **0.84** | **0.78** | **0.81** |
| | mPLUG-Owl2 | VCD | × | 0.76 | 0.74 | 0.81 | 0.77 |
| | | | ✓ | **0.77** | **0.77** | 0.76 | **0.77** |
| | | OPERA | × | 0.71 | 0.66 | 0.88 | 0.75 |
| | | | ✓ | **0.73** | **0.68** | **0.89** | **0.77** |

Table 22: Evaluation results of POPE on GQA across different settings.

| Setting | Model | Method | w/DSCR | Accuracy | Precision | Recall | F1-Score |
|---|---|---|---|---|---|---|---|
| Random | LLaVA-1.5 | VCD | × | 0.88 | 0.87 | 0.90 | 0.88 |
| | | | ✓ | **0.88** | **0.88** | **0.89** | **0.89** |
| | | OPERA | × | 0.88 | 0.85 | 0.89 | 0.88 |
| | | | ✓ | **0.87** | **0.87** | **0.88** | **0.88** |
| | Qwen-VL | VCD | × | 0.85 | 0.84 | 0.87 | 0.85 |
| | | | ✓ | **0.85** | **0.84** | **0.86** | **0.85** |
| | | OPERA | × | 0.84 | 0.84 | 0.86 | 0.84 |
| | | | ✓ | **0.84** | **0.85** | **0.86** | **0.85** |
| | mPLUG-Owl2 | VCD | × | 0.85 | 0.91 | 0.77 | 0.83 |
| | | | ✓ | **0.84** | **0.93** | 0.73 | **0.82** |
| | | OPERA | × | 0.84 | 0.92 | 0.76 | 0.84 |
| | | | ✓ | **0.84** | **0.93** | **0.74** | **0.84** |
| Popular | LLaVA-1.5 | VCD | × | 0.85 | 0.81 | 0.90 | 0.85 |
| | | | ✓ | **0.84** | **0.82** | **0.89** | **0.85** |
| | | OPERA | × | 0.84 | 0.78 | 0.89 | 0.84 |
| | | | ✓ | **0.84** | **0.79** | **0.89** | **0.84** |
| | Qwen-VL | VCD | × | 0.77 | 0.72 | 0.87 | 0.79 |
| | | | ✓ | **0.76** | **0.72** | **0.86** | **0.78** |
| | | OPERA | × | 0.78 | 0.72 | 0.86 | 0.78 |
| | | | ✓ | **0.77** | **0.73** | **0.86** | **0.78** |
| | mPLUG-Owl2 | VCD | × | 0.79 | 0.80 | 0.77 | 0.79 |
| | | | ✓ | **0.79** | **0.83** | 0.73 | **0.78** |
| | | OPERA | × | 0.79 | 0.81 | 0.76 | 0.79 |
| | | | ✓ | **0.79** | **0.83** | **0.74** | **0.79** |
| Adversarial | LLaVA-1.5 | VCD | × | 0.80 | 0.75 | **0.90** | 0.82 |
| | | | ✓ | 0.80 | **0.76** | 0.88 | 0.82 |
| | | OPERA | × | 0.80 | 0.75 | **0.90** | 0.82 |
| | | | ✓ | 0.80 | **0.76** | 0.89 | 0.82 |
| | Qwen-VL | VCD | × | 0.77 | 0.73 | 0.86 | 0.79 |
| | | | ✓ | 0.77 | 0.73 | 0.86 | 0.79 |
| | | OPERA | × | **0.78** | 0.72 | 0.86 | 0.79 |
| | | | ✓ | 0.77 | **0.73** | 0.86 | 0.79 |
| | mPLUG-Owl2 | VCD | × | 0.77 | 0.77 | **0.77** | **0.77** |
| | | | ✓ | 0.77 | **0.80** | 0.73 | 0.76 |
| | | OPERA | × | 0.77 | 0.79 | **0.77** | 0.76 |
| | | | ✓ | 0.77 | **0.80** | 0.74 | 0.76 |

