# OpenReview forum: "Mitigating Hallucination in Vision-Language Model with Depth and Spatial-aware Key-Value Refinement"
_ICLR.cc/2026/Conference — ICLR 2026 Poster_

### Official Review · Reviewer_gacf · 2025-10-17

**Soundness:** 3
**Presentation:** 3
**Contribution:** 3
**Rating:** 6
**Confidence:** 5

**Summary:**

This paper proposes a training-free and lightweight method to mitigate LVLM hallucinations by encoding spatial and depth priors. The core idea is to re-weight the key–value pairs of visual tokens in the cache to enhance grounding, without additional training.

**Strengths:**

1. This paper is well-motivated and supported by insightful visualizations. The method is novel and achieves consistent improvements on both standard object hallucination benchmarks and attribute/spatial hallucination benchmarks.
2. The paper is overall well written and easy to follow.

**Weaknesses:**

1. Since both MME and POPE are yes/no questions, it would strengthen the evaluation to extend Table 3 to include detailed CHAIR scores across LVLMs and baselines.
2. The paper provides spatial and depth priors but does not provide enough discussion of related work on grounding visual information.

**Clarification**
1. (Major) In Table 1, Qwen-VL Count row: your method is not the best (+155) but is still bolded. Similarly, for Qwen-VL Poster row, your method (165.99) is not the best. Please correct this.
2. In Figure 1, the Non-Hallucination subfigure is a bit confusing. why are there two “A. Yes” labels?

**Questions:**

1. I saw the ablation results in Tables 10–14. Could you provide more intuition on why key-only reweighting performs better, and why selecting layers 10–39 (deep layers) works best?
2. To confirm: my understanding is that you apply re-weighting only to the visual token cache, not the text tokens. If that’s correct, it would help to make this explicit throughout sec 2.3.
3. A broader question: do you envision future LVLMs incorporating 3D vision encoders directly to enhance spatial reasoning?

would be happy to raise my score if my concerns are addressed

---

> ### Author Response · Authors · 2025-11-21
> **1. “Since both MME and POPE are yes/no questions, it would strengthen the evaluation to extend Table 3 to include detailed CHAIR scores across LVLMs and baselines.”**
>
> **Response:**
> We thank the reviewer for this helpful suggestion. We agree that yes/no benchmarks such as MME and POPE should be complemented with caption-based hallucination metrics.
>
> In the current version, Table 3 already reports CHAIR scores for other methods and baselines. In the revision, we extend this analysis by adding CHAIR results for mPLUG-Owl2 in a new table (Table 10). As shown in Table 10, applying DSCR to the same mPLUG-Owl2 backbone improves both CHAIR\_S and CHAIR\_I. This is consistent with the trends in Table 3 for other methods and supports the claim that DSCR reduces object hallucination both in yes/no tasks and in captioning settings.
>
> *Table 3: Evaluation results on the CHAIR dataset using LLaVA-1.5.*
>
> | Method   | CHAIR$_\text{S}$ ↓ | CHAIR$_\text{I}$ ↓ | Recall ↑ | Avg. Len. |
> |----------|:------------------:|:------------------:|:--------:|:---------:|
> | Baseline | 52.2               | 14.3               | 78.2     | 101.4     |
> | VCD      | 52.6               | 15.5               | 76.4     | 103.0     |
> | OPERA    | 48.6               | 13.6               | 78.5     | 97.6      |
> | **DSCR** | **37.6**           | **11.6**           | **79.5** | **96.8**  |
>
> *Table 10: Evaluation results on the CHAIR dataset using mPLUG-Owl2.*
>
> | Method   | CHAIR\_s | CHAIR\_l | Recall | Avg. Len |
> |----------|:--------:|:--------:|:------:|:--------:|
> | Baseline | 74.6     | 35.7     | 53.6   | 100.2    |
> | VCD      | 66.4     | 21.6     | 73.8   | 109.0    |
> | OPERA    | 62.8     | 21.0     | 71.3   | 108.8    |
> | **DSCR** | **57.8** | **18.3** | **76.7** | **107.8** |

---

> ### Author Response · Authors · 2025-11-21
> **2. “The paper provides spatial and depth priors but does not provide enough discussion of related work on grounding visual information.”**
>
> **Response:**
> We thank the reviewer for pointing this out. In the revised manuscript, we add a dedicated subsection *“Visual Grounding in Vision–Language Models”* to the Related Work (Section B.3). This subsection discusses recent grounding approaches such as SpatialVLM [1], ReVisiT [2], GroundVLP [3], and Mono3DVG [4], and clarifies how DSCR differs from and complements these methods. In particular, these works mainly improve grounding through additional training or explicit grounding heads, whereas DSCR refines the visual KV cache at inference time without retraining.
>
> [1] Chen, Boyuan, et al. “SpatialVLM: Endowing Vision-Language Models with Spatial Reasoning Capabilities.” *Proceedings of the IEEE/CVF Conference on Computer Vision and Pattern Recognition (CVPR)*, 2024.
>
> [2] Cho, Beomsik, and Jaehyung Kim. “Revisit What You See: Disclose Language Prior in Vision Tokens for Efficient Guided Decoding of LVLMs.” *arXiv preprint* arXiv:2506.09522, 2025.
>
> [3] Shen, Haozhan, et al. “GroundVLP: Harnessing Zero-shot Visual Grounding from Vision-Language Pre-training and Open-Vocabulary Object Detection.” *arXiv preprint* arXiv:2312.15043, 2023.
>
> [4] Zhan, Yang, Yuan Yuan, and Zhitong Xiong. “Mono3DVG: 3D Visual Grounding in Monocular Images.” *arXiv preprint* arXiv:2312.08022, 2023.

---

> ### Author Response · Authors · 2025-11-21
> **3. “In Table 1, Qwen-VL Count row: your method is not the best (+155) but is still bolded. Similarly, for Qwen-VL Poster row, your method (165.99) is not the best. Please correct this.”**
>
> **Response:**
> We thank the reviewer for carefully catching this formatting mistake. In the revised manuscript, we correct the boldface in Table 1 so that only the best scores are highlighted. DSCR entries are no longer bolded in rows where they are not the top performer.

---

> ### Author Response · Authors · 2025-11-21
> **4. “In Figure 1, the Non-Hallucination subfigure is a bit confusing. why are there two ‘A. Yes’ labels?”**
>
> **Response:**
> We thank the reviewer for pointing out this confusion. We have revised Figure 1 so that non hallucination examples are no longer labeled with two identical “A. Yes” strings. Each answer is now displayed in a clearly distinguishable way, with separate markers for each method, which removes the ambiguity.

---

> ### Author Response · Authors · 2025-11-21
> **5. “I saw the ablation results in Tables 10–14. Could you provide more intuition on why key-only reweighting performs better, and why selecting layers 10–39 (deep layers) works best?”**
>
> **Response:**
>
> We thank the reviewer for this question. We summarize the intuition in two parts.
>
> **Why key-only reweighting is more stable:**
> Attention scores depend on inner products between queries and keys. DSCR improves grounding by changing keys so that tokens within the same object become more aligned and tokens across boundaries less similar. This directly reshapes where attention concentrates. Value vectors, on the other hand, encode semantic content with statistics that subsequent layers expect. Changing values without retraining can disturb these statistics and introduce instability. Our ablations show that key-only refinement gives consistent improvements across models and datasets, while key-plus-value helps in some cases but is less stable.
>
> **Why layers 10 to 39 work best:**
> According to previous interpretation studies on ViTs and vision–language transformers, early layers mainly capture low-level cues such as color, texture, and edges, while middle and deeper layers gradually move toward object- and part-level representations and stronger alignment with text tokens [1,2]. Applying DSCR in this middle to deep layer range (layers 10–39) allows us to improve the visual key structure where object-level grounding takes place, without interfering with the very early feature extraction stages. Appendix Figure 10 in the revised manuscript provides a layer-wise PCA visualization for a fixed example and illustrates this pattern for both hallucination and non-hallucination cases.
>
> [1] Fan, Haoqi, et al. *Multiscale Vision Transformers.* Proceedings of the IEEE/CVF International Conference on Computer Vision (ICCV), 2021.
> [2] Neo, Clement, et al. *Towards Interpreting Visual Information Processing in Vision-Language Models.* Proceedings of the International Conference on Learning Representations (ICLR), 2025.

---

> ### Author Response · Authors · 2025-11-21
> **6. "To confirm: my understanding is that you apply re-weighting only to the visual token cache, not the text tokens. If that’s correct, it would help to make this explicit throughout sec 2.3.”**
>
> **Response:**
> We thank the reviewer for checking this detail. Your understanding is correct. DSCR applies reweighting only to KV cache entries associated with visual tokens. The KV cache for text tokens, including the system prompt and user question, is unchanged. In the revised manuscript, we have updated Section 2.3 (highlighted in blue) to state this explicitly and remove any ambiguity.

---

> ### Author Response · Authors · 2025-11-21
> **7. “A broader question: do you envision future LVLMs incorporating 3D vision encoders directly to enhance spatial reasoning?”**
>
> **Response:**
> We thank the reviewer for raising this forward-looking question. We agree that integrating 3D vision encoders into future LVLMs is a promising direction for strengthening spatial reasoning. Three-dimensional cues such as monocular depth, multi-view geometry, and point cloud features can provide geometric structure that is difficult to obtain from 2D images alone. We also believe that DSCR can play a complementary role to such 3D-aware LVLMs, and we are grateful for this valuable research direction.

---

> ### Author Response · Authors · 2025-11-25
>
> Dear Reviewer gacf,
>
> Thank you for your thoughtful review and constructive feedback. We have carefully addressed all of your concerns and questions in our official author responses and incorporated the corresponding revisions in the manuscript. We would be grateful if you could kindly review our replies and confirm whether they sufficiently resolve your concerns.
>
> Sincerely,
> The Authors

---

> > ### Comment · Reviewer_gacf · 2025-11-25
> >
> > Hi,
> >
> > Most of my concerns have been addressed. I'd love to raise my score to 8.

---

> ### Author Response · Authors · 2025-11-25
> **Thank you**
>
> Dear Reviewer gacf,
>
> Thank you for your kind follow up. We truly appreciate your careful reading and constructive feedback. We are glad that our revisions addressed your concerns, and we are grateful for your thoughtful suggestions, which helped us improve the quality and clarity of the paper.
>
> Thank you again for your time and support.
>
> Sincerely,
> The Authors

---

### Official Review · Reviewer_UhZY · 2025-10-30

**Soundness:** 2
**Presentation:** 3
**Contribution:** 3
**Rating:** 4
**Confidence:** 4

**Summary:**

This paper studies the representational origin of visual hallucinations in vision–language models (VLMs) and proposes a training-free method, Depth and Spatial-aware Cache Refinement (DSCR), that modifies the visual key–value (KV) cache before decoding. The central empirical observation is that successful grounding correlates with coherent alignment of neighboring key vectors, while hallucination correlates with isotropically scattered keys that blur object boundaries. DSCR injects depth cues and 2D spatial proximity into the KV cache by forming a proximity-weighted refinement of keys (optionally values), which clusters vectors within objects and separates vectors across surfaces. The method is model- and query-agnostic, adds negligible overhead, and shows gains on multiple hallucination benchmarks. The paper also introduces a hallucination benchmark to occlusions and similar-depth confounders.

**Strengths:**

- Clear mechanistic story grounded in the model’s internal representations. The paper ties hallucination to a loss of neighboring-key coherence, and supports this with PCA-based visualizations across layers and attention diagnostics that show increased attention to image tokens when DSCR is applied.
- Simple, training-free intervention with broad applicability. DSCR only modifies the KV cache at inference, without changing weights. The reported overhead is small and the method improves several VLM backbones.
- Consistent gains on hallucination evaluations. Across MME, POPE/RePOPE, CHAIR, and AMBER, the method shows improvements; the depth hallucination mini-benchmark highlights gains in occluded and similar-depth cases.

**Weaknesses:**

- The link between “neighboring-key coherence” and hallucination is mainly supported by PCA visualizations and attention trends. The paper describes that, in hallucination cases, keys scatter and object boundaries blur, but it does not define a quantitative measure of “neighboring-key similarity” or “key-vector dispersion,” nor does it report large-scale correlations with hallucination/error rates. To strengthen the claim, it is benificial to introduce simple, layer-wise metrics (for example, average cosine similarity with spatial neighbors, or a local PCA explained-variance ratio) and report correlation/predictive power with confidence intervals.

- Comparisons to alternative geometry-injection routes are missing. Since DSCR effectively injects geometric priors, it would be informative to compare against baselines that (i) use a stronger vision encoder that already has geometric priors or (ii) concat depth features with visual tokens as inputs for VLM without KV rewriting. This would clarify the unique benefits of operating in KV cache.

- Evidence for depth–spatial complementarity is limited. In Table 13, depth-only and spatial-only achieve the same total score (645), while the combined setting is only modestly higher (650), with improvements concentrated in one submetric. Figure 4(c,d) validates each component on its targeted subset but does not establish that combining both consistently outperforms either alone across the same benchmarks. More side-by-side results with multiple runs, error bars, and paired tests, and stratified by scene attributes (e.g., occlusion density, depth discontinuities, similar-depth distractors), are helpful to show that the combination truly helps.

- Breadth beyond hallucination-centric tasks is limited. The positive COCO captioning result is helpful, but a broader suite (e.g., additional VQA settings) would better establish that there is no negative transfer to general VL capabilities.

**Questions:**

- Since DSCR leaves VLM weights frozen, could the lack of adaptation to DSCR-refined keys limit the attainable gains? Have you tried enabling fine-tuning with DSCR active, and if so, did you observe larger improvements or any shift in preferred settings (e.g., Key-only vs. Key+Value, layer ranges)?
- DSCR modifies only the visual branch of VLMs. Can it mitigate hallucinations that arise when the language model misinterprets visual tokens (i.e., unsatisfied vision-language alignment)? Do you have controlled analyses or case studies indicating whether DSCR helps in such language model-driven error modes?
- Semantic richness vs. key scattering. The paper describes hallucination cases where keys scatter and object boundaries blur. Could similar patterns also appear in images with genuinely rich, heterogeneous semantics? After applying DSCR, is there any measurable loss of semantic diversity that could harm performance on VL tasks requiring fine-grained distinctions or rare attributes?
- Minor presentation. In Figure 1(a), the non-hallucination examples show “X” and “O” even though there is no wrong answer, which is confusing. Also, the “A.” prefix for answers can be misread as an option label. Clarifying the notation would improve readability.

---

> ### Author Response · Authors · 2025-11-21
> **1. "The paper claims a link between neighboring-key coherence and hallucination, but only shows PCA and attention trends and does not provide quantitative metrics"**
>
> **Response:**
> We thank the reviewer for highlighting the need for a quantitative measure of neighboring-key similarity and its correlation with hallucination to complement the PCA plots and attention trends. In the revised manuscript, we complement the PCA plots with a quantitative metric, called Boundary Contrast (BC), and show that this metric is strongly related to hallucination behavior.
>
> First, we define BC as follows. Given a depth map $d_p$ on the $24 \times 24$ token grid and four-connected neighbors $N(p)$, we first compute a simple depth gradient
> $G_p = \max_{q \in N(p)} \lvert d_p - d_q \rvert$.
> Tokens with large $G_p$ (for example, above the 90th percentile per image) are selected as boundary tokens $B$. For each boundary token $p \in B$, we split its neighbors into inner and outer sets based on the depth difference. Using $\ell_2$-normalized keys $k_p^{(\ell)}$ at layer $\ell$, we define
> $\mathrm{BC}{\text{in}} = \mathbb{E}{p \in B},\mathbb{E}{q \in N{\text{in}}(p)} \cos\bigl(k_p^{(\ell)}, k_q^{(\ell)}\bigr)$,
> $\mathrm{BC}{\text{out}} = \mathbb{E}{p \in B},\mathbb{E}{q \in N{\text{out}}(p)} \cos\bigl(k_p^{(\ell)}, k_q^{(\ell)}\bigr)$,
> and $\mathrm{BC} = \mathrm{BC}{\text{in}} - \mathrm{BC}{\text{out}}$.
>
> A large BC means that key vectors are much more similar on the same depth side of a boundary than on the other side. This corresponds to sharp object separation. When boundary structure is weak, $\mathrm{BC}{\text{in}}$ and $\mathrm{BC}{\text{out}}$ become similar and BC is close to zero.
>
> Next, we evaluate $(\mathrm{BC}{\text{in}}, \mathrm{BC}{\text{out}}, \mathrm{BC})$ for the baseline VLM and the DSCR-augmented model on the MME hallucination subsets and our Depth Hallucination Mini Benchmark. For clarity, we divide the examples into four groups:
>
> 1. **Group 1:** baseline wrong, DSCR correct
> 2. **Group 2:** both wrong
> 3. **Group 3:** both correct
> 4. **Group 4:** baseline correct, DSCR wrong
>
> The main observation is that applying DSCR yields the largest increase in BC for Group 1. A larger BC means that key vectors of tokens on the same depth side of a boundary become more similar. At the same time, their similarity to tokens on the opposite side becomes lower. As a result, object boundaries are more clearly separated in the representation space. In contrast, when hallucination occurs, BC becomes small and tokens on both sides of the boundary tend to be mixed. Based on this observation, the BC analysis provides quantitative support for our PCA-based qualitative finding that non-hallucination cases form clear object-level clusters, whereas hallucination cases show mixed boundary tokens. Detailed values and explanations are provided in Section E.10 and Table 18.
>
> *Table 18. Boundary Contrast statistics on the MME hallucination subsets and the Depth Hallucination Mini-Benchmark. For the LLaVA-1.5 model, DSCR increases BC_in and decreases BC_out on hallucination-fix cases, which leads to a larger overall BC compared to the baseline.*
>
> | Split                         | Method   | BC_in | BC_out | BC   |
> |------------------------------|----------|:-----:|:------:|:----:|
> | Baseline wrong, DSCR correct | Baseline | 0.71  | 0.64   | 0.07 |
> |                              | DSCR     | 0.87  | 0.57   | 0.30 |
> | Both wrong                   | Baseline | 0.69  | 0.63   | 0.06 |
> |                              | DSCR     | 0.73  | 0.60   | 0.13 |
> | Both correct                 | Baseline | 0.82  | 0.61   | 0.21 |
> |                              | DSCR     | 0.88  | 0.60   | 0.28 |
> | Baseline correct, DSCR wrong | Baseline | n/a   | n/a    | n/a  |
> |                              | DSCR     | n/a   | n/a    | n/a  |

---

> ### Author Response · Authors · 2025-11-21
> **2. “Since DSCR injects geometric priors, the paper should compare against alternatives such as (i) stronger vision encoders with built-in geometry and (ii) simply concatenating depth features to visual tokens instead of KV rewriting.”**
>
> **Response:**
> We thank the reviewer for suggesting these two alternative geometry-injection routes. Using (i) a stronger vision encoder with geometric priors and (ii) concatenating depth features with visual tokens are both promising ways to further reduce hallucination. However, both directions come with practical issues that need to be addressed.
>
> First, when using a geometry-aware vision encoder, it typically needs to be pre-trained or fine-tuned with additional 3D or depth supervision, and often further fine-tuned to align with the language branch. This introduces non-trivial costs in data preparation, training, and optimization. It can also lead to an encoder that is closely tailored to a particular LVLM architecture and may lose some generalization ability.
>
> Second, simply concatenating depth features to visual tokens usually requires additional fine-tuning in practice. LVLMs are trained with fixed input dimensions and token lengths, so adding depth channels or depth tokens increases the sequence length and the KV cache size, which in turn increases memory usage and latency.
>
> If these issues are carefully handled, both directions become interesting and complementary extensions to DSCR. Exploring DSCR together with geometry-specialized vision encoders or lightweight depth–token fusion modules is a promising topic that we plan to investigate in future work.

---

> ### Author Response · Authors · 2025-11-21
> **3. "Figure 4(c,d) validates each component on its targeted subset but does not establish that combining both consistently outperforms either alone across the same benchmarks.”**
>
> **Response:**
> We apologize for the confusion caused by Figure 4(c,d). The current labeling can indeed give the impression that each curve corresponds to a single component. In fact, all curves in Figure 4(c,d) show the performance of the full DSCR method evaluated on different subsets, and we do not present per-component results in this figure. In the revised manuscript, we have corrected the labels and caption of Figure 4(c,d) to clearly indicate that the plotted curves correspond to DSCR, not to individual components. We hope this clarification helps avoid misinterpretation, and we kindly ask the reviewer to refer to the updated figure in the revision.

---

> ### Author Response · Authors · 2025-11-21
> **4. “Breadth beyond hallucination-centric tasks is limited. The positive COCO captioning result is helpful, but a broader suite (e.g., additional VQA settings) would better establish that there is no negative transfer to general VL capabilities.”**
>
> **Response:**
> To examine whether DSCR causes negative transfer to general vision–language capabilities, we evaluate it on two standard non-hallucination benchmarks: COCO captioning and VQAv2.
>
> On the non-hallucination COCO captioning task, as reported in Table 6, DSCR improves BLEU-4 from 0.122 to 0.235, CIDEr from 0.529 to 0.909, and SPICE from 0.162 to 0.193. This indicates that DSCR significantly strengthens visual grounding without harming overall caption quality or diversity.
>
> On the VQAv2 benchmark, Table 11 shows small improvements in overall accuracy: LLaVA-1.5 increases from 79.40 to 79.67, and Qwen2.5-VL increases from 84.27 to 84.40 when DSCR is applied. These results show that DSCR does not hurt general VL performance and can even provide a modest benefit beyond hallucination-centric benchmarks.
>
> *Table 6: COCO image captioning performance with and without DSCR.*
>
> | Method    | BLEU-4 | CIDEr | SPICE |
> |----------|:------:|:-----:|:-----:|
> | Baseline | 0.122  | 0.529 | 0.162 |
> | **DSCR** | **0.235** | **0.909** | **0.193** |
>
> *Table 11: Evaluation results on the VQAv2 dataset.*
>
> | Metric           | LLaVA-1.5 Baseline | LLaVA-1.5 DSCR | Qwen2.5-VL Baseline | Qwen2.5-VL DSCR |
> |------------------|:------------------:|:--------------:|:-------------------:|:---------------:|
> | Overall Accuracy | 79.40              | **79.67**      | 84.27               | **84.40**       |

---

> ### Author Response · Authors · 2025-11-21
> **5. "Since DSCR leaves VLM weights frozen, could the lack of adaptation to DSCR-refined keys limit the attainable gains? Have you tried enabling fine-tuning with DSCR active, and if so, did you observe larger improvements or any shift in preferred settings?”**
>
> **Response:**
> We thank the reviewer for suggesting the possibility of combining DSCR with fine-tuning. We consider allowing the model to adapt to DSCR-refined keys a natural and promising extension, and we are grateful for this valuable research direction. In this work we focus on the frozen-weight setting, and we plan to explore such fine-tuning based extensions in future work.

---

> ### Author Response · Authors · 2025-11-21
> **6. "DSCR modifies only the visual branch of VLMs. Can it mitigate hallucinations that arise when the language model misinterprets visual tokens? Do you have controlled analyses or case studies indicating whether DSCR helps in such language model-driven error modes?”**
>
> **Response:**
>
> We thank the reviewer for raising this point. DSCR is not originally designed to directly resolve hallucinations that arise purely from the language model itself.
>
> In our experiments, DSCR mainly targets image-based hallucinations that occur when the model incorrectly groups or localizes objects in the scene. In such cases, sharpening object boundaries in the visual key space can help the existing cross-attention align the query more easily with the correct objects.
>
> In the current version, as the reviewer pointed out, we do not include a separate controlled analysis or case study that focuses only on error modes driven by the language model. The benchmarks we use mostly consist of questions where visual grounding is crucial, and within the scope of our inspection, cases that can be explained purely as language-model errors appear relatively rare. Systematically collecting and categorizing such errors driven by the language model, and analyzing how they interact with DSCR, is an important and meaningful direction that we plan to pursue in future work.

---

> ### Author Response · Authors · 2025-11-21
> **7.	“Semantic richness vs. key scattering. "**
>
> **Response:**
> We thank the reviewer for raising this point. DSCR is not designed to reduce semantic richness. It does not change what the visual encoder stores in each token. It only adjusts the key geometry so that tokens from the same object are closer and tokens on different depth sides of a boundary are more separated.
>
> Even in images with rich and heterogeneous semantics, the tokens already carry diverse content. DSCR keeps this content and aims to make it easier for attention to use by sharpening object boundaries, rather than collapsing different meanings into a single cluster.
>
> Empirically, we do not observe a loss of semantic diversity. On COCO captioning, which requires fine grained descriptions and rare attributes, DSCR improves BLEU-4, CIDEr, and SPICE compared to the baseline (Table 6). This suggests that DSCR preserves fine grained information instead of flattening it, and does not harm vision–language tasks that rely on subtle or rare attributes.
>
> *Table 6: COCO image captioning performance with and without DSCR*
> | Method    | BLEU-4 | CIDEr | SPICE |
> |----------|:------:|:-----:|:-----:|
> | Baseline | 0.122  | 0.529 | 0.162 |
> | **DSCR** | **0.235** | **0.909** | **0.193** |

---

> ### Author Response · Authors · 2025-11-21
> **8. “In Figure 1(a), the non-hallucination examples show ‘X’ and ‘O’ even though there is no wrong answer, which is confusing. Also, the ‘A.’ prefix for answers can be misread as an option label.”**
>
> Response:
> We thank the reviewer for pointing out this confusing presentation. In the revised manuscript, we update Figure 1(a) so that non hallucination examples are shown only with clear colored markers for each method. We remove the “X/O” notation and the “A.” prefix that could be mistaken for option labels. This improves readability and avoids ambiguity about which answer belongs to which method.

---

> ### Author Response · Authors · 2025-11-25
>
> Dear Reviewer UhZY,
>
> Thank you again for your careful review and constructive feedback. We have now addressed all of your listed weaknesses, questions, and minor presentation comments in the official author responses (including the added points 1–8). We would be grateful if you could kindly take a moment to review our replies and confirm whether they sufficiently resolve your concerns.
>
> Sincerely,
> The Authors

---

> ### Author Response · Authors · 2025-11-27
> **Reminder for Reviewer UhZY**
>
> Dear Reviewer UhZY,
>
> Thank you again for your careful review and constructive feedback. We have already provided responses to all of your listed points and questions in the official author replies and incorporated the corresponding updates in the revised manuscript. We would be very grateful if you could kindly take a moment to review our replies when convenient and let us know if any concerns remain.
>
> Sincerely,
> The Authors

---

> ### Author Response · Authors · 2025-11-27
>
> Dear Reviewer UhZY,
>
> Thank you again for your careful review and constructive feedback.
>
> This is a gentle reminder that the discussion period closes on Dec 3. We have already submitted detailed responses to all of your points in the official author replies, and the revised manuscript reflects these updates.
>
> For context, after reviewing our rebuttal and follow up clarifications, Reviewer qhDk and Reviewer gacf both raised their scores from 6 to 8, and Reviewer nKMD maintained the score of 8 with a positive assessment of our revisions. We would be very grateful if you could also take a moment to review our replies when convenient. If you still see any remaining issues or have further suggestions, please feel free to let us know. We are happy to clarify or improve accordingly within the remaining discussion window.
>
> Sincerely,
> The Authors

---

### Official Review · Reviewer_qhDk · 2025-10-30

**Soundness:** 3
**Presentation:** 3
**Contribution:** 2
**Rating:** 6
**Confidence:** 4

**Summary:**

The paper introduces Depth and Spatial-aware Cache Refinement (DSCR) — a lightweight, training-free method to suppress visual hallucinations in LVLMs. The authors identify that hallucinations stem from incoherent alignment among key vectors (KVs) within the Transformer’s attention mechanism, which disrupts the cross-modal grounding between visual and textual inputs. DSCR addresses this by refining the Transformer’s KV cache using geometric and spatial priors derived from monocular depth estimation. Through integrating 3D depth cues and 2D spatial proximity, DSCR enforces coherence among tokens representing the same object and separates tokens across different surfaces, thereby improving visual grounding and reducing false object generation. Extensive experiments over multiple benchmarks demonstrate up to 23% accuracy improvement.

**Strengths:**

1. The paper provides a novel and elegant perspective on the origin of hallucinations by analyzing the internal coherence of key vectors within multimodal Transformers.

2. The DSCR method is training-free, model-agnostic, and computationally efficient.

3. The experimental evaluation is comprehensive and rigorous, and the writing is clear.

**Weaknesses:**

1. As shown on the right side of Figure 5, while DSCR leverages pre-computed depth maps, its performance inevitably depends on the quality of depth estimation, which may introduce inaccuracies in complex lighting or occlusion scenarios.

2. Why would the misalignment of key vectors weaken the model’s ability to correctly interpret visual inputs? I believe there is no essential connection. I hope the authors can provide further clarification.

3. This paper focuses on optimizing the static reasoning phase, without exploring joint training or adaptive tuning of deep cues. I believe that the misalignment between visual and textual modality features is also an important factor contributing to hallucination. Should additional training be incorporated into DSCR to enhance alignment with textual vectors?

**Questions:**

1. Does DSCR take the user’s input query into account? If it only considers image features while ignoring textual features, will this lead to performance degradation in handling more complex queries?

2. Why does an isotropic distribution of key vectors tend to produce hallucinations?

---

> ### Author Response · Authors · 2025-11-21
> **1. “As shown on the right side of Figure 5, while DSCR leverages pre-computed depth maps, its performance inevitably depends on the quality of depth estimation, which may introduce inaccuracies in complex lighting or occlusion scenarios.”**
>
> **Response:**
> Regarding the concern about the reliance on depth estimation quality, we evaluate how sensitive DSCR is to the choice and quality of the depth estimator. Our results show that DSCR is robust to realistic depth estimation errors and does not require highly accurate depth maps.
> In Tables 8 and 12, DSCR consistently improves over the baseline even when we use compact or noisy monocular depth estimators. When we replace the depth model with different alternatives, key metrics vary only within a few percentage points, and the hallucination mitigation effect remains stable even with lightweight models. These results suggest that coarse depth already provides a useful prior that helps the model focus on relevant visual evidence, and that DSCR does not rely on precise depth estimation to be effective.
>
> *Table 8: GPU memory, per-image inference time, and performance comparisons of DSCR using different depth estimators.*
> | Depth Model                          | GPU (MiB) | Time (sec/img) |   OCR  | Color | Count | Existence | Position | Posters |
> |--------------------------------------|:---------:|:--------------:|:------:|:-----:|:-----:|:---------:|:--------:|:-------:|
> | Depth-Anything-v2~\citep{yang2024depth} | 2134.1    |      1.34      | 132.5  | 175.0 | 160.0 |  195.0    | 120.0    | 140.48  |
> | MiDaS-Lite~\citep{ranftl2020midas}      | 1264.2    |      1.15      | 125.0  | 180.0 | 160.0 |  195.0    | 111.67   | 132.65  |
> | DPT-Lite~\citep{ranftl2021dpt}          |  526.1    |      1.54      | 125.0  | 180.0 | 160.0 |  195.0    | 111.67   | 132.65  |
>
>
> *Table 12: Ablation study for varying the size of depth estimation model, conducted on the MME dataset using LLaVA-1.5 model.*
> | Model              | Type (Size)    | Existence | Count   | Position | Color   | Total    |
> |--------------------|----------------|:---------:|:-------:|:--------:|:-------:|:--------:|
> | -                  | -              | 173.33    | 116.66  | 113.33   | 123.33  | 526.66   |
> | Depth-Anything-v2  | Small (0.03B)  | 190.00    | 153.33  | 120.00   | 170.00  | 633.33   |
> | Depth-Anything-v2  | Base (0.1B)    | 190.00    | 160.00  | 120.00   | 170.00  | 640.00   |
> | **Depth-Anything-v2**  | **Large (0.3B)** | **195.00** | **160.00** | **120.00** | **175.00** | **650.00** |

---

> ### Author Response · Authors · 2025-11-21
> **2. “Why would the misalignment of key vectors weaken the model’s ability to correctly interpret visual inputs? I believe there is no essential connection. I hope the authors can provide further clarification.”**
>
> **Response:**
> We would like to clarify the connection between key vector misalignment and the model’s ability to interpret visual inputs. This point can be explained from three perspectives: theoretical, qualitative, and quantitative.
>
> - **(1) Theoretical explanation (scaled dot product attention).**
>   In scaled dot product attention, the weight on visual token $j$ for a query token $q_i$ is proportional to
>   $\exp\bigl(q_i^\top k_j / \sqrt{d}\bigr)$.
>   When key vectors for tokens on the same object are well aligned and clearly separated from background tokens, the inner product $q_i^\top k_j$ tends to be larger on that object than on other locations. The softmax then concentrates on the object tokens, and the query aggregates mostly relevant visual evidence.
>   When key vectors are scattered and many of them are also similar to background directions, many tokens receive similar inner products. The softmax becomes flatter and attention spreads to both relevant and irrelevant regions. The representation for the query mixes signal for the target object with unrelated information, which effectively lowers the signal to noise ratio of the attended visual features.
>
> - **(2) Qualitative evidence (PCA of key vectors).**
>   In our PCA visualizations, non hallucination cases show compact clusters that follow object regions and display clear boundaries between objects. Hallucination cases show scattered key vectors where boundary tokens from different objects are intermixed and object separation becomes weak. This qualitative behavior matches the attention based argument above. When object boundaries are not well represented in key space, attention is more likely to blend multiple objects and produce incorrect visual interpretations.
>
> - **(3) Quantitative evidence (Boundary Contrast, BC).**
>   In the revised manuscript, we further introduce the Boundary Contrast (BC) metric in Section E.10. BC uses depth gradients to approximate object boundaries, selects boundary tokens, and compares cosine similarity to depth consistent neighbors on the same side and to neighbors on the other depth side. A large BC means that key vectors change sharply at estimated object boundaries and are coherent within each side.
>   On examples where the baseline hallucinates but DSCR produces the correct answer (Group 1), BC increases noticeably after applying DSCR. On examples where both the baseline and DSCR are already correct, BC changes only slightly. These results show that weak boundary contrast and misaligned neighboring keys are closely related to hallucination, and that DSCR improves this structure mainly on hallucination cases rather than globally distorting the representation.

---

> ### Author Response · Authors · 2025-11-21
> **3. "Should additional training be incorporated into DSCR when visual and textual modality features are misaligned?"**
>
> **Response:**
> We thank the reviewer for suggesting the use of additional training to further improve visual–text alignment and reduce hallucination. We agree that lightweight training-based extensions on top of DSCR are a promising direction. For example, one could attach a small adapter or projection layer and fine-tune it so that DSCR-refined visual features are better aligned with textual representations, while keeping the main encoders largely fixed. We view such training-based extensions as complementary to our method and plan to explore them in future work.

---

> ### Author Response · Authors · 2025-11-21
> **4. "Does DSCR take the user’s input query into account? If it only considers image features while ignoring textual features, will this lead to performance degradation in handling more complex queries?”**
>
> **Response:**
> We respond to the reviewer’s concern that ignoring textual features might harm performance on complex queries. DSCR does not use the user’s input query in its refinement step, and our experiments show that it does not degrade performance even for complex, visually grounded queries.
>
> In our method, DSCR refines only the KV cache of visual tokens, while the textual KV cache and all model parameters remain unchanged. During refinement, depth and spatial proximity are used so that tokens from the same object become more coherent and tokens on different surfaces become better separated. After this step, the frozen cross attention operates on these refined visual features, so subsequent queries are handled in the same way regardless of how simple or complex they are.
>
> In this sense, DSCR does not tailor the representation to a specific query. Instead, it makes object boundaries clearer in the visual features and improves how the model understands the image itself, which is beneficial for a wide range of queries, including complex ones.

---

> ### Author Response · Authors · 2025-11-21
> **5. “Why does an isotropic distribution of key vectors tend to produce hallucinations?”**
>
> **Response:**
> We thank the reviewer for asking for a more detailed explanation of this point. An isotropic key distribution means that key vectors of different patches are scattered in many directions without clear object aligned structure.
>
> From the attention perspective:
>
> - Attention scores depend on inner products $q^\top k_i$.
> - If many keys point in random directions, these inner products become similar in magnitude for many tokens.
> - The softmax then produces a flat attention distribution spread over many locations.
>
> When attention is flat, relevant and irrelevant patches contribute with similar weight, which weakens visual evidence. The model then leans more on language priors and common textual patterns. This increases the chance of generating objects or attributes that are not in the image, that is, hallucinations.
>
> Our PCA plots show that non hallucination cases have structured clusters along object boundaries, while hallucination cases show scattered keys and blurred boundaries. Boundary Contrast is low in the latter and increases when DSCR corrects the answer. Together, these observations support the claim that isotropic key distributions promote flat attention and thereby contribute to hallucinations. DSCR mitigates this by restoring structure in key space so that attention can focus on the correct regions.

---

> > ### Comment · Reviewer_qhDk · 2025-11-24
> >
> > Thank you to the authors for their thorough response. Although the model has a relatively large GPU footprint, its performance is strong and the theoretical foundation is well-developed. The authors have provided reasonable explanations that have resolved my concerns. Therefore, I have decided to raise my score.

---

> > > ### Author Response · Authors · 2025-11-24
> > > **Thank you**
> > >
> > > Dear Reviewer qhDk,
> > >
> > > Thank you very much for your thoughtful review and for your follow-up comment. We are very grateful that our revisions and clarifications have addressed your concerns and that you decided to raise your score from 6 to 8. Your detailed questions on the robustness to depth estimation quality, the role of key vector alignment, and the interaction between visual and textual modalities were extremely helpful in sharpening both our analysis and presentation. We will carefully incorporate these clarifications and insights into the final version of the paper.
> > >
> > > Thank you again for your time, constructive feedback, and support.
> > >
> > > Best regards,
> > > Authors of Submission 7218

---

### Official Review · Reviewer_nKMD · 2025-10-31

**Soundness:** 4
**Presentation:** 3
**Contribution:** 4
**Rating:** 8
**Confidence:** 4

**Summary:**

This paper attempts to mitigate hallucination in VLMs by first studying the key vector distribution in model's transformer layers. The authors discovered that key vector distributions exhibit distinct object boundary patterns when models are faithful, while showing blurred object borders when hallucinating. This phenomena motivated the authors to propose DSCR, a training-free method that corrects key and value vectors using depth and spatial similarity maps as guidance. Under DSCR, a separate depth estimator model is used to generate a depth map. Coupled with spatial proximity, weightage is calculated to rebalance the key and value vectors during inference. Experimental results show that DSCR achieves high performance on various hallucination benchmarks. In addition, DSCR is also complementary with other hallucination mitigation methods, further boosting the evaluation performance on multiple open source VLM models.

The paper contributes to the research community by introducing a new correlation between key vector similarity and hallucination. The training free method can be applied on top of many existing works to further reduce VLM hallucination and improve model performance.

**Strengths:**

Overall, the paper is well written and the concepts are easy to follow. The authors also offer a new direction to study the underlying cause of hallucination. The proposed DSCR method shows good performance and generalisability. It is an efficient design that is complementary with many existing hallucination mitigation methods. The validity of DSCR design is sufficiently supported by many experiments. The author conducted extensive analysis experiments including attention score value comparison and key vector distribution visualisation for models with and without DSCR correction. The authors also conducted comprehensive ablation experiments such as depth only, spatial only and depth with spatial weightage calculation as shown in appendix.

**Weaknesses:**

It is insightful for the authors to reveal the different key vector PCA visualisations for hallucinating and non-hallucinating VLM inference scenarios. However, this qualitative analysis can be sensitive to different factors such as object size, object position, difficulty in text query, etc. This slightly undermines the reliability of this discovery and thus the motivation of the method design. The authors could provide more evidence, such as quantitative experimental results, to show that the phenomena is universal, that blurring of object boundaries key vectors is common for different images and query types.

**Questions:**

1. In figure 3 different shades represent different values. However, why does the distribution pattern look like this? Which 2 image patches are compared?
2. Based on the ablation experiment table 14, it shows that the best performance is achieved when DSCR is applied to layer 10–39. Why does the final method apply to all layers instead of following the finding from this ablation study?
3. For the key vector PCA visualisation in figure 1, does the pattern stay the same across different layers? Or is the distinct object boundary only specific to certain attention layers?

---

> ### Author Response · Authors · 2025-11-21
> **1. “Please provide quantitative experimental evidence showing how key vector alignments at object boundaries impact VLM hallucinations.”**
>
> **Response:**
> We appreciate the reviewer’s insightful comment. In the revised manuscript, we complement the PCA plots with a quantitative metric, called Boundary Contrast (BC), and show that this metric is strongly related to hallucination behavior.
>
> First, we define BC as follows. Given a depth map $d_p$ on the $24 \times 24$ token grid and four-connected neighbors $N(p)$, we first compute a simple depth gradient $G_p = \max_{q \in N(p)} \lvert d_p - d_q \rvert$. Tokens with large $G_p$ (for example, above the 90th percentile per image) are selected as boundary tokens $B$. For each boundary token $p \in B$, we split its neighbors into inner and outer sets based on the depth difference. Using $\ell_2$-normalized keys $k_p^{(\ell)}$ at layer $\ell$, we define
> $\mathrm{BC}{\text{in}} = \mathbb{E}{p \in B},\mathbb{E}{q \in N{\text{in}}(p)} \cos\bigl(k_p^{(\ell)}, k_q^{(\ell)}\bigr)$,
> $\mathrm{BC}{\text{out}} = \mathbb{E}{p \in B},\mathbb{E}{q \in N{\text{out}}(p)} \cos\bigl(k_p^{(\ell)}, k_q^{(\ell)}\bigr)$,
> and $\mathrm{BC} = \mathrm{BC}{\text{in}} - \mathrm{BC}{\text{out}}$.
>
> A large BC means that key vectors are much more similar on the same depth side of a boundary than on the other side. This corresponds to sharp object separation. When boundary structure is weak, $\mathrm{BC}{\text{in}}$ and $\mathrm{BC}{\text{out}}$ become similar and BC is close to zero.
>
> Next, we evaluate $(\mathrm{BC}{\text{in}}, \mathrm{BC}{\text{out}}, \mathrm{BC})$ for the baseline VLM and the DSCR-augmented model on the MME hallucination subsets and our Depth Hallucination Mini Benchmark. For clarity, we divide the examples into four groups:
>
> 1. **Group 1:** baseline wrong, DSCR correct
> 2. **Group 2:** both wrong
> 3. **Group 3:** both correct
> 4. **Group 4:** baseline correct, DSCR wrong
>
> The main observation is that applying DSCR yields the largest increase in BC for Group 1.A larger BC means that key vectors of tokens on the same depth side of a boundary become more similar. At the same time, their similarity to tokens on the opposite side becomes lower. As a result, object boundaries are more clearly separated in the representation space. In contrast, when hallucination occurs, BC becomes small and tokens on both sides of the boundary tend to be mixed. Based on this observation, the BC analysis provides quantitative support for our PCA-based qualitative finding that non-hallucination cases form clear object-level clusters, whereas hallucination cases show mixed boundary tokens. Detailed values and explanations are provided in Section E.10 and Table 18.
>
>
> *Table 18. Boundary Contrast statistics on the MME hallucination subsets and the Depth Hallucination Mini-Benchmark. For the LLaVA-1.5 model, DSCR increases BC_in and decreases BC_out on hallucination-fix cases, which leads to a larger overall BC compared to the baseline.*
>
> | Split                         | Method   | BC_in | BC_out | BC   |
> |------------------------------|----------|:-----:|:------:|:----:|
> | Baseline wrong, DSCR correct | Baseline | 0.71  | 0.64   | 0.07 |
> |                              | DSCR     | 0.87  | 0.57   | 0.30 |
> | Both wrong                   | Baseline | 0.69  | 0.63   | 0.06 |
> |                              | DSCR     | 0.73  | 0.60   | 0.13 |
> | Both correct                 | Baseline | 0.82  | 0.61   | 0.21 |
> |                              | DSCR     | 0.88  | 0.60   | 0.28 |
> | Baseline correct, DSCR wrong | Baseline | n/a   | n/a    | n/a  |
> |                              | DSCR     | n/a   | n/a    | n/a  |

---

> ### Author Response · Authors · 2025-11-21
> **2. "In figure 3 different shades represent different values. However, why does the distribution pattern look like this? Which 2 image patches are compared?”**
>
> **Response:**
> We appreciate the reviewer’s comment. Here we clarify what Figure 3 represents and why the distribution pattern looks like that.
>
> Figure 3 shows a pairwise proximity matrix over patches from a single image. Each row index $i$ and column index $j$ correspond to image patches, and the entry $(i, j)$ represents the proximity between patch $i$ and patch $j$. Darker shades indicate higher proximity. The main diagonal is darkest because each patch is compared with itself and this gives the maximum value.
>
> The pattern of shades depends on how we define proximity:
>
> - **Depth-based proximity.**
>   Patches that belong to the same object tend to have similar depth. When the 2D patch grid is flattened into a sequence, these patches appear with nearby indices and form dark blocks near the diagonal. Patches from different objects have larger depth differences and appear in lighter regions.
>
> - **Spatial proximity.**
>   Proximity is computed from the Euclidean distance between patch centers in the image plane. Patches are ordered in a fixed sequence (for example, raster scan order). This produces a strong main diagonal and several secondary diagonals that correspond to spatial neighbors at fixed index offsets.
>
> In all cases, both indices $i$ and $j$ run over patches from the same image. Thus, each matrix in Figure 3 visualizes depth-based or spatial proximity between all pairs of patches in that image.

---

> ### Author Response · Authors · 2025-11-21
> **3. “Based on the ablation experiment table 14, it shows that the best performance is achieved when DSCR is applied to layer 10–39. Why does the final method apply to all layers instead of following the finding from this ablation study?”**
>
> **Response:**
> We appreciate the reviewer’s careful reading. Our final method does not apply DSCR to all layers. In all main experiments, we follow the ablation in Table 14 and apply DSCR only to layers 10 to 39. This layer range is explicitly specified in Section C.4 of the appendix, and we will make sure this is easy to locate for readers.

---

> ### Author Response · Authors · 2025-11-21
> **4. “For the key vector PCA visualisation in figure 1, does the pattern stay the same across different layers? Or is the distinct object boundary only specific to certain attention layers?”**
>
> **Response:**
> We appreciate the reviewer’s question. The distinct object boundary pattern in the key PCA plots is most visible in middle and higher visual layers, and it is not restricted to a single attention layer.
>
> Several interpretation studies on ViTs and vision-language transformers report a similar trend [1,2]. Early layers mainly respond to low-level cues such as color, texture, and edges. Middle and higher layers gradually move toward object- or part-level representations and show stronger alignment with text tokens. Our observations are consistent with these results.
>
> In the revised manuscript, we provide a layer-wise PCA analysis for a fixed representative example (new Figure 10 in the appendix). We present both hallucination and non-hallucination cases and compare the key distributions before and after applying DSCR. For non-hallucination cases, the baseline model already produces object-aligned clusters and clear boundary structure from middle layers onward, and DSCR preserves this pattern. For hallucination cases, the same layers of the baseline show almost isotropic key distributions in which boundary tokens from different objects are mixed, whereas DSCR recovers separated clusters and sharper boundaries in several successive layers.
>
> These results support that the behavior illustrated in Figure 1 reflects a common property of middle and higher layers, rather than an example from a carefully chosen single layer.
>
> [1] Fan, Haoqi, et al. *Multiscale Vision Transformers*. Proceedings of the IEEE/CVF International Conference on Computer Vision (ICCV), 2021.
> [2] Neo, Clement, et al. *Towards Interpreting Visual Information Processing in Vision-Language Models*. Proceedings of the International Conference on Learning Representations (ICLR), 2025.

---

> ### Comment · Reviewer_nKMD · 2025-11-24
> **Response to authors' rebuttal**
>
> I appreciate the authors' effort in addressing my questions and further improving this work. I will maintain my initial rating of 8 as I think this is a good poster paper, though it may not meet the threshold for an oral or spotlight presentation.

---

> > ### Author Response · Authors · 2025-11-24
> > **Thank you**
> >
> > Dear Reviewer nKMD
> >
> > We sincerely thank Reviewer nKMD for the careful evaluation, constructive questions, and positive overall assessment of our work. We are also grateful that you chose to maintain your rating of 8.
> >
> > In the revised version, we have incorporated your suggestions on adding quantitative evidence for key vector behavior near object boundaries, clarifying the proximity visualizations, specifying the layer selection strategy for DSCR, and providing a layer-wise PCA analysis. These changes have, in our view, strengthened the clarity and soundness of the paper and improved both the empirical support and the presentation. We greatly appreciate your valuable feedback.

---

### Author Response · Authors · 2025-11-21
**COMMON RESPONSE**

**COMMON RESPONSE**

We sincerely thank Reviewers nKMD, qhDk, UhZY, and gacf for their careful evaluation and constructive feedback. We are encouraged that they highlighted the following strengths of our work:

- **New perspective on hallucination mechanism:** Reviewers appreciated that our analysis links hallucinations to the coherence of neighboring key vectors and to object boundary patterns in Transformer representations, which provides a clear and novel direction to study the representational origin of hallucinations in VLMs. (Reviewer nKMD; Reviewer qhDk; Reviewer UhZY)

- **Training free, model agnostic, lightweight design:** DSCR was recognized as a simple and efficient method that refines the KV cache at inference without additional training, can be applied to diverse LVLM backbones, and can be combined with many existing hallucination mitigation methods. (Reviewer nKMD; Reviewer qhDk; Reviewer UhZY; Reviewer gacf)

- **Strong and consistent empirical gains:** Reviewers noted that DSCR achieves excellent performance and consistent improvements across multiple hallucination benchmarks and architectures, covering both standard object hallucination and attribute or spatial hallucination settings. Reviewer qhDk in particular highlighted that our experiments report accuracy gains of up to 23%. (Reviewer nKMD; Reviewer qhDk; Reviewer UhZY; Reviewer gacf)

- **Comprehensive analysis and ablations:** The paper was recognized for extensive analyses such as attention score comparisons and PCA visualizations of key vector distributions with and without DSCR, as well as ablation studies on depth only, spatial only, combined depth plus spatial weighting, and layer ranges. These results support the current design and also motivated several useful suggestions for additional quantitative metrics and comparisons, which we address in our response. (Reviewer nKMD; Reviewer UhZY; Reviewer gacf)

- **Clear and accessible presentation:** Multiple reviewers commented that the paper is well written, easy to follow, and supported by effective figures and visualizations that clarify both the method and its empirical effects. (Reviewer nKMD; Reviewer qhDk; Reviewer UhZY; Reviewer gacf)

We are grateful for these positive assessments and for the detailed suggestions in the weaknesses and questions. In the remainder of our response, we address each concern in turn and provide additional experiments and analyses to further support the effectiveness and robustness of DSCR.

**All modifications in the main paper are highlighted in blue.**

---

### Author Response · Authors · 2025-12-03
**Summary of Reviewer Feedback and Discussion Outcomes**

## **Key strengths highlighted by reviewers**

- **New perspective on hallucination mechanism**
  Reviewers appreciated that we link hallucinations to the coherence of neighboring key vectors and to object boundary patterns in Transformer representations, which gives a concrete way to study the representational origin of hallucination in VLMs.
  *(Reviewers nKMD, qhDk, UhZY)*

- **Training free, model agnostic, lightweight design**
  DSCR refines only the visual KV cache at inference, requires no additional training, and can be plugged into diverse LVLM backbones and combined with other hallucination mitigation methods.
  *(Reviewers nKMD, qhDk, UhZY, gacf)*

- **Strong and consistent empirical gains**
  Reviewers noted that DSCR achieves strong and consistent improvements across multiple hallucination benchmarks, including MME, POPE, RePOPE, CHAIR, AMBER, and a depth sensitive benchmark, with gains up to 23 percent.
  *(Reviewers nKMD, qhDk, UhZY, gacf)*

- **Comprehensive analyses and ablations**
  The paper includes attention diagnostics, PCA visualizations of key distributions with and without DSCR, ablations on depth only and spatial only variants, layer ranges, and depth model size, which together support the method design.
  *(Reviewers nKMD, UhZY, gacf)*

- **Clear and accessible presentation**
  Multiple reviewers commented that the paper is well written and easy to follow, with figures and visualizations that make the method and its effects understandable.
  *(Reviewers nKMD, qhDk, UhZY, gacf)*


## **Main questions and our responses**

- **Quantitative evidence for neighboring key coherence and hallucination**
  Reviewers asked for quantitative metrics beyond PCA and attention plots. We introduced the Boundary Contrast (BC) metric, which measures how sharply key vectors change across depth based boundaries, and evaluated it on MME hallucination subsets and our Depth Hallucination Mini Benchmark. DSCR significantly increases BC on examples where it corrects hallucinations while leaving already correct cases nearly unchanged, which supports a strong link between boundary aligned key structure and hallucination behavior.
  *(Reviewers nKMD, UhZY)*

- **Sensitivity to depth estimation quality**
  Reviewers were concerned that DSCR depends on depth estimation accuracy. We added experiments with multiple monocular depth estimators of different sizes and qualities and showed that DSCR consistently improves over the baseline in all cases. Performance varies only within a few points across depth models, which indicates that coarse depth is sufficient and DSCR is robust to realistic depth noise.
  *(Reviewer qhDk)*

- **Why key alignment and isotropy matter for hallucination**
  Reviewers requested a clearer explanation of why misaligned or isotropically scattered keys hurt grounding. We clarified this from an attention perspective and with BC. When keys are aligned within objects and separated across boundaries, attention concentrates on relevant regions. When keys scatter and BC is small, attention becomes flat over many patches, which weakens visual evidence and makes the model rely more on language priors, leading to hallucinations. DSCR restores structure in key space so that attention focuses on the correct regions.
  *(Reviewer qhDk)*

- **Layer range and key only refinement**
  Reviewers asked why key only reweighting and layers 10 to 39 work best. We explained that attention weights depend only on queries and keys, so adjusting keys directly controls where attention concentrates, while modifying values without retraining can disturb semantic statistics. Prior interpretation work shows that middle and deeper layers carry object level and text aligned features, which matches our finding that applying DSCR on layers 10 to 39 yields the best trade off between effectiveness and stability.
  *(Reviewers nKMD, gacf)*

- **Interaction with textual queries and general VL performance**
  Reviewers questioned whether ignoring the user query in the refinement step could harm complex queries and general VL capabilities. We clarified that DSCR refines only the visual KV cache and leaves textual KV and all parameters frozen. We added non hallucination evaluations on COCO captioning and VQAv2, where DSCR improves BLEU, CIDEr, SPICE, and slightly improves VQAv2 accuracy, which shows that DSCR strengthens visual grounding without negative transfer to general VL performance.
  *(Reviewers qhDk, UhZY)*

---

> ### Author Response · Authors · 2025-12-03
>
> - **Alternative ways to inject geometric priors**
>   Reviewers suggested comparisons with stronger geometry aware vision encoders and simple depth concatenation to visual tokens. We discussed that these directions typically require additional pre training or fine tuning, increase sequence length and KV size, and can be architecture specific. We positioned DSCR as a complementary, training free alternative that operates entirely in the KV cache, and we highlighted joint designs with geometry specialized encoders or lightweight depth fusion as future work.
>   *(Reviewer UhZY)*
>
> - **Evaluation breadth and related work on grounding**
>   Reviewers asked for broader caption based hallucination metrics and more discussion of visual grounding literature. We extended CHAIR evaluations to another backbone, mPLUG Owl2, where DSCR again yields the lowest hallucination rates, and we added a dedicated related work subsection on visual grounding methods and clarified how DSCR complements training based grounding approaches. We also fixed minor issues, including boldface errors in Table 1 and confusing labels in Figure 1.
>   *(Reviewer UhZY, Reviewer gacf)*
>
>
> ## **Score changes and discussion outcome**
>
> After our rebuttal and added analyses, **Reviewer qhDk raised the score from 6 to 8 on Nov 23, 2025**, and **Reviewer gacf raised the score from 6 to 8 on Nov 25, 2025**.
> **Reviewer nKMD maintained the initial positive rating of 8** and posted a follow-up comment on Nov 24, 2025, explicitly endorsing the work as a good poster paper.
> Reviewer UhZY did not update the score during the discussion period, but our responses addressed all listed weaknesses, questions, and minor presentation issues.
>
> We are grateful to all reviewers for their careful reading and constructive feedback, and we appreciate the area chair’s time and effort in assessing our submission. We are happy to provide any further clarification if needed.

---

### Meta-Review · Area_Chair_TheS · 2026-01-07

**Summary:**

This paper proposes DSCR, a training-free inference-time method to reduce hallucinations in VLMs by modifying the visual KV cache in transformer layers. The authors’ key observation is representational: when the model is faithful, key vectors form object-aligned clusters with sharp boundaries; when it hallucinates, key vectors become more isotropic and boundaries blur. They report consistent improvements on multiple hallucination benchmarks and show DSCR composes well with other mitigation methods.

All reviewers appreciate this paper's contribution and the authors provided a solid rebuttal that help accept this paper.

**Reviewer Concerns:**

Most of the reviewers concerns are addressed during the rebuttal, which is acknowledged reasonably in the reviewers' response

**Reviewer Scores:**

Two reviewers have acknowledged to raise their score from 6 to 8. The only negative reviewer 4 has received a comprehensive rebuttal so they will very likely to update their score.

---

### Decision · Program_Chairs · 2026-01-26

Accept (Poster)